# Harmonious genetic combinations rewire regulatory networks and flip gene essentiality

Aaron M. New [1] & Ben Lehner [1,2,3]

We lack an understanding of how the full range of genetic variants that occur in individuals can interact. To address this shortcoming, here we combine diverse mutations between genes in a model regulatory network, the galactose (GAL) switch of budding yeast. The effects of thousands of pairs of mutations fall into a limited number of phenotypic classes. While these effects are mostly predictable using simple rules that capture the 'stereotypical' genetic interactions of the network, some double mutants have unexpected outcomes including constituting alternative functional switches. Each of these 'harmonious' genetic combinations exhibits altered dependency on other regulatory genes. These cases illustrate how both pairwise and higher epistasis determines gene essentiality and how combinations of mutations rewire regulatory networks. Together, our results provide an overview of how broad spectra of mutations interact, how these interactions can be predicted, and how diverse genetic solutions can achieve 'wild-type' phenotypic behavior.

[1] Centre for Genomic Regulation (CRG), The Barcelona Institute for Science and Technology, Dr. Aiguader 88, 08003 Barcelona, Spain. [2] Universitat Pompeu Fabra (UPF), Barcelona, Spain. [3] Institució Catalana de Recerca i Estudis Avançats (ICREA), 08010 Barcelona, Spain. Correspondence and requests for materials should be addressed to B.L. (email: ben.lehner@crg.eu)

Human genomes contain millions of genetic variants. Each of these variants can have diverse effects, for example quantitatively increasing, decreasing, or changing the activity of individual genes[1]. Understanding and predicting how the particular combination of variants present in each individual affects molecular processes and phenotypic traits is a fundamental challenge for human genetics and evolutionary biology[2]. To date, however, systematic analyses of how mutations in different genes combine to influence phenotypes have used only one or a few mutations in each gene, most often an inactivating null allele[3,4]. A more complete understanding of how mutations combine in individuals will require functionally diverse mutations in individual genes to be combined in large numbers[5].

The GAL regulatory (GALR) system from yeast is a promising model to begin such studies because it is mechanistically well-understood, has relatively few molecular players, and is an important model of gene network function and evolution[6,7] (Fig. 1a). This network is required for sensing the sugar galactose and then inducing transport (via Gal2) and the Leloir pathway proteins Gal1p, Gal7p, and Gal10p necessary to metabolize this sugar as a carbon source for growth. The core of this network consists of three regulatory genes and their protein products: GAL4, a transcriptional activator; GAL80, its repressor; and GAL3, which acts as a GAL sensor by inhibiting Gal80p as an activated Gal3p-Galactose-ATP complex[8] (Fig. 1a). The GAL1 locus, encoding the first Leloir enzyme galactokinase (GALK) Gal1p, is a paralog of GAL3 that can also effect galactose sensing. The ancestral GALK locus likely encoded a bifunctional protein that was duplicated to give rise to the main sensor Gal3p and the kinase Gal1p[9].

Here, we present a systematic analysis of how mutations in GALR genes with diverse individual effects combine to alter gene expression and growth phenotypes. We quantify the effects of >5000 pairs of mutations and find that, despite strong genetic interactions in the network, they typically exhibit a few stereotypical phenotypic behaviors, allowing their outcomes to be predicted. However, for individual genotypes, unexpected genetic interactions can be important. In particular, interactions between gain-of-function mutations in the GAL4 and GAL80 regulators can reconstitute wild-type (WT)-like switches, exhibiting multiple phenotypes of the WT GAL system, including repression of the pathway in glucose, and induction of the pathway and robust growth in the presence of galactose. We show that these "harmonious" combinations do not simply reconstruct the original wild-type network but rather represent alternative, re-wired switches in which the functions and importance of the galactose sensors GAL1 and GAL3 have changed. This includes a case where the essentiality of GAL1 and GAL3 are reversed. This illustrates how the selective pressure on a pair of paralogous genes can change substantially as a result of higher-order epistasis with mutations in other genes. Taken together, our results illustrate how gene essentiality and function can be flipped by interactions between mutations in other genes.

## Results

### Systematically combining mutations in GALR genes.
To systematically explore how mutations in the GALR genes GAL4, GAL3, and GAL80 affect GAL pathway output we first tested all combinations of GALR WT or coding sequence deletions (Δ) to give $2^3 = 8$ unique genotypes (Fig. 1b, c and Methods). GAL pathway activation and growth were quantified using flow cytometry to measure cell density and the expression of a Gal1p-YeCitrine (YFP) reporter in two environments. The first environment was an initial "uninducing" condition where cells had reached saturation in media with a low concentration of glucose,

and the second condition was "inducing" after 12 h of growth in galactose. As expected, across these eight genotypes, the GAL pathway exhibits three phenotypic classes: Inducible, Uninducible, and Constitutive (Fig. 1d–f, Supplementary Fig 1, and Source Data). The WT GALR system is Inducible because it is repressed in glucose and activated in galactose. The single mutants ΔGAL3 and ΔGAL4 result in Uninducible phenotypes because they do not activate GAL expression in any conditions, and ΔGAL80 drives Constitutive phenotypes because GAL expression is activated in glucose. The triple mutant and all double-mutant genotypes yielded Uninducible phenotypes except the ΔGAL80 + ΔGAL3 mutant, which is Constitutive because WT GAL4 is not repressed in the ΔGAL80 background. GAL1-YFP expression was highly predictive of growth rate across these genetic backgrounds (Fig. 1f).

### Generating a diverse set of GALR alleles.
Next, to investigate how richer spectra of mutations in the three-GALR genes interact, we used PCR mutagenesis to generate variants of the three-GALR genes and phenotyped them using flow cytometry (Supplementary Tables 1–3 and Methods). We observed GALR alleles that individually behaved as detrimental, mildly detrimental, and WT-like. We also observed mutations in GAL80 ("GAL80S") that led to Uninducible phenotypes, as well as mutations in GAL4 ("GAL4C") that caused Constitutive expression. Since these were relatively rare, we generated a number of previously described gain-of-function mutants in GAL80[10,11] and we mutagenized position L868 of GAL4, a site in the Gal80p-binding interface of the Gal4p activation domain where mutation to P had previously been shown to drive a constitutive phenotype[12] (Methods and Supplementary Tables 1–11).

### Quantifying the effects of >5000 double mutants.
We selected a subset of these mutants based on their expression in glucose and galactose representing these different phenotypic classes, including strong loss-of-function alleles, gain-of-function alleles, and phenotypically WT-like variants (Methods). We used these alleles to perform a pairwise combinatorially complete experiment where all pairs of GALR variants were combined together and the resulting phenotype quantified. In total, we combined 46 alleles of GAL3, 39 of GAL80 and 43 of GAL4. After transformation, this library included 98% of the possible pairwise genotypes, giving 5317 unique double-mutant combinations (Fig. 2a and Methods). We quantified GAL pathway expression and growth characteristics for each genotype as before (Supplementary Fig 1 and Source Data).

The resulting phenotypes were more diverse than when combining gene deletions (Fig. 2b and Supplementary Fig. 2), but most genotypes still grouped into a limited number of phenotypic classes (Fig. 2c, Supplementary Fig. 3A–D, and Methods), with 92% of the 5317 genotypes falling into the Inducible, Constitutive and Uninducible expression classes observed when combining gene deletions (Fig. 2c). A further ~5% of "Leaky" genotypes were closest to Inducible profiles but in glucose exhibited a detectable fraction of ON cells with low mean expression (Supplementary Fig. 3). Finally, 3% of samples fell into a "Weak expression" class with low maximal expression in glucose and galactose. Neither of these behaviors was observed when combining null alleles.

### Double mutants exhibit limited expression phenotypes.
We found that only a limited number of expression phenotype classes were observed for double mutants when single mutants from particular phenotypic classes were combined (Fig. 2d, e). For example, when Leaky GAL80 mutants were combined with

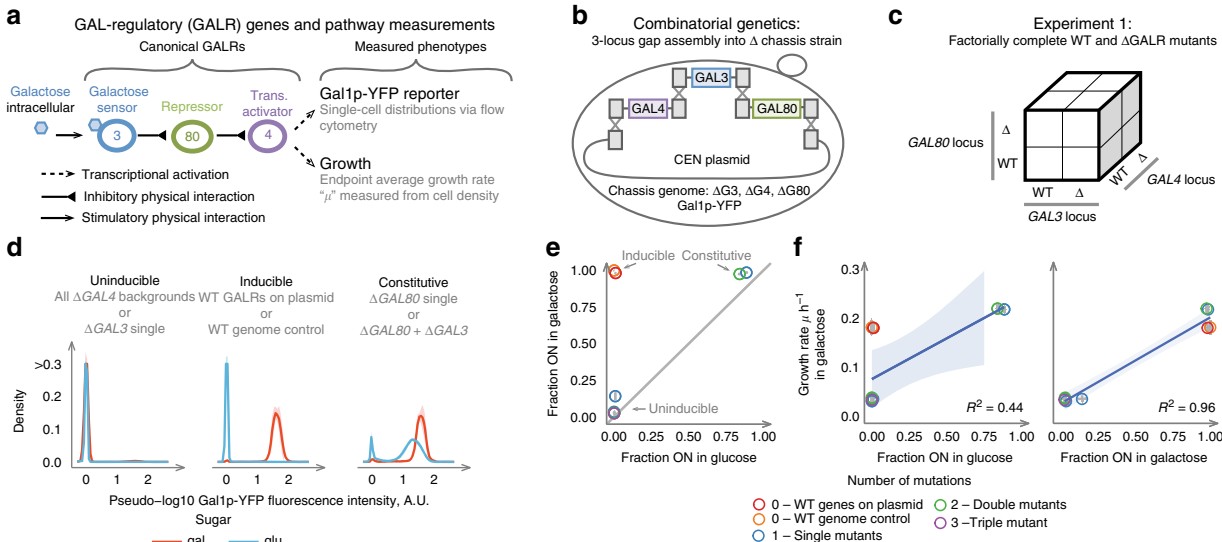

**Fig. 1** Combinatorial genetic analysis of the *GAL* pathway. **a** Overview of *GAL* pathway regulation. **b** In vivo homologous recombination to combine alleles of three GALR genes into a chassis strain with all three genes deleted. **c** Cubic representation of combinatorially complete genetics experiments combining alleles in multiple loci. **d** Three phenotypic classes of GALR deletion mutants. Lines and shading are the mean and ± 1 SD of genotypes falling into the given expression class ($N = 4$ for $N = 2$ independent transformations for each unique genotype). Blue lines are samples growing in glucose and red lines are samples after 12 h of growth in galactose. **e** Expression distributions summarized by the fraction of cells ON in galactose and glucose. Points are mean values for each unique genotype, and are colored to reflect the number of mutations. **f** Fraction ON in glucose and galactose predicts growth rate. Blue line and shaded region are the expectation and 95% confidence interval of a linear model

Uninducible *GAL3* mutants, 108/113 of the double mutants (96%) fell into the Leaky phenotypic class, with the remaining 4% classified as Inducible. The single mutants in *GAL3*, *GAL4*, and *GAL80* fell into two, four, and four phenotypic classes, respectively, giving a total of 32 distinct paired-locus expression class combinations in the double mutants. For 30/32 combinations (94%), >90% of the resulting double mutants fell into a single phenotypic class (Fig. 2d and Supplementary Table 4). The most diverse double-mutant phenotypes were observed when combining Uninducible *GAL80* mutants with Constitutive *GAL4* mutants, in which case all five phenotypic classes were observed. However, even in this case, 56% (20/36) of the double mutants were Constitutive, a significant enrichment ($p = 1.5 \times 10^{-4}$, Chi-squared test and Fig. 2d and Supplementary Fig. 4). These results suggest that gene expression in the GAL system changes in a largely stereotypical manner when combining mutations in the GALR genes, especially for loss-of-function mutations.

**Abundant epistasis when combining GALR mutations**. We next asked whether the growth rate in galactose of double mutants could be predicted from the phenotypes of single mutants. We first tested the extent to which the growth rates of double mutants were predicted by simply multiplying the relative growth rates of the single mutants. This is the most commonly used null model for how mutations combine in both quantitative genetics and functional genomics and assumes that mutations have independent effects on growth[3,13]. The multiplicative model of single mutants explained 55% of the variance in the growth rate of the 5120 double-mutants (Supplementary Fig. 5A, Methods). There is therefore substantial epistasis when combining pairs of mutations in the GALR genes. Predictive performance varied widely across different combinations of single-mutant classes with some combinations being generally poorly predicted. For example, all pairwise combinations of Constitutive *GAL80* and Uninducible *GAL3* were Constitutive and therefore grew at high rates, when the prediction was that they would grow slowly due to *GAL3*'s low growth rate. Similarly, fast-growing

Constitutive *GAL4* variants were predicted to grow slowly in Uninducible *GAL3* backgrounds, when the double mutant remained Constitutive.

**Single mutants predict the fitness of double mutants**. We next asked whether knowing the expression class of each single mutant allowed more accurate prediction of the growth rates of the double mutants. Specifically, we tested a model in which all the mutations in a particular gene in a given gene expression phenotypic class combine with all the mutations in a second gene in a given expression phenotypic class to give the same double-mutant growth phenotype. Using five gene expression classifications for each single mutant (Inducible, Constitutive, Uninducible, Leaky, and Weak Expression) gives a total of 23 double-mutant classes in the dataset (e.g., Uninducible *GAL3* x Constitutive *GAL80*). We simply used the mean growth rate of all of the genotypes falling into each of the 23 double-mutant classes as the prediction of the growth rate of each double mutant in the class. So, for example, the mean of all double mutants of an Uninducible *GAL3* plus a Constitutive *GAL80* was used as the predicted growth rate for all double mutants whose single *GAL3* variant was Inducible and whose single *GAL80* variant was Constitutive. These mean values explained 89%, 90%, 91% of total growth rate variance for double mutants in *GAL3* vs. *GAL4*, *GAL80* vs. *GAL3*, and *GAL80* vs. *GAL4* pairings, respectively (Methods). Across all double mutants, this model explained 91% of growth rate variance.

Using a coarser classification of only three single-mutant expression phenotypes (Inducible, Constitutive, and Uninducible) gives 14 possible double-mutant classes. Using the mean growth rate in each of these 14 classes to predict the growth of all double mutants still explains 89% of growth rate variance (Fig. 2f, Supplementary Fig. 5B, C, D, Methods, and Source Data). Moreover, even if we only used the mean growth rate of one random genotype from each expression class pairing, the predictions remain accurate, with a median of 86% (IQR = 84.0–88.0%) and 88% (IQR = 86.3–89.6%) of the total growth rate variance explained in the remaining >5000 double mutants

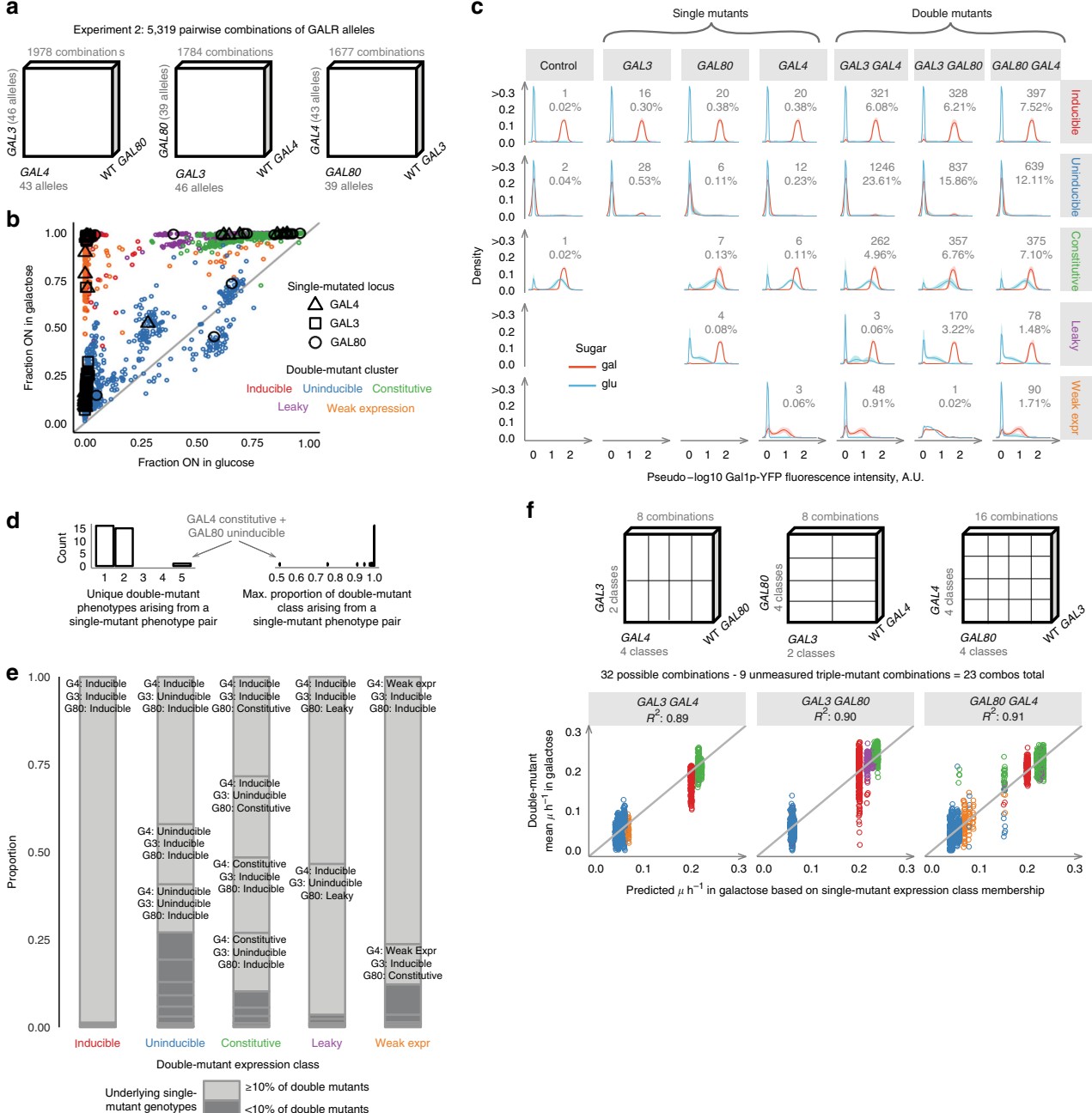

**Fig. 2** Phenotypic variation from pairwise GALR allele combinations. **a** Alleles obtained by mutagenic PCR of a given GALR locus were paired with alleles of the other two loci. **b** Single and double mutants have more diverse phenotypes than gene deletions. Points represent the means of $N = 2$ to $N = 36$ (median $N = 2$) independent replicates across $N = 1$ to $N = 18$ independent transformants (median $N = 1$). Large black points are single mutants, with shape variation corresponding to the locus of the mutation, and small points are double mutants colored according to their resulting double-mutant phenotype. Gray line is $X = Y$. **c** Five classes of expression distributions. Lines and shaded regions are density means and ± 1 SD across genotypes of each class and genotype. Integers and percentages indicate the number of genotypes in each panel. **d** The count of unique double-mutant phenotype classes arising from single-mutant expression class combinations (left), and for each of these combinations, the proportion of pairings, which fell into the most prevalent class (right). **e** The single-mutant combinations, leading to a given double-mutant phenotype shown as stacked bar plots. Single-mutant combinations that comprised >10% of genotypes, leading to the indicated double-mutant phenotype are labeled on the chart. **f** The 23 mean growth rates corresponding to all pairwise combinations of single-mutant expression classes (horizontal axis) were used to predict 5152 double-mutant growth rates (vertical axis). (Methods)

for the three and five expression class models, respectively. Double-mutants arising from pairings between Constitutive GAL4 alleles with Uninducible GAL80 alleles were the least predictable in their resulting growth rates (Fig. 2d, points with growth predictions at ~0.15/h) and gene expression classes (Supplementary Fig. 4). Knowing the identity of the mutated

genes is, however, useful for making accurate predictions: three- and five-class expression models that ignore the identity of the mutated genes, respectively, explain 77% and 79% of the growth rate variance, with large errors for particular class combinations (Supplementary Fig. 5E, Methods). Thus, despite the strong epistasis in this system, simple models that capture the main

"stereotypical" genetic interactions between loci based on their gene expression can accurately predict how pairs of diverse mutations interact.

**"Harmonious" allele combinations make functional switches**. Although most double mutants exhibited highly stereotyped gene expression and predictable fitness patterns, particular combinations of mutations sometimes had very different phenotypes compared to those expected from their individual phenotypic effects. For example, 12% of pairings of Constitutive GAL4 alleles and Uninducible GAL80 variants yielded Inducible phenotypes, examples of genetic suppression (Supplementary Fig. 4). Similar interactions have been previously described for distinct GAL4-GAL80 allele combinations[10,12,14]. Other suppressor combinations yielding WT-like Inducible or Leaky phenotypes included all pairings of Uninducible GAL3 with Leaky GAL80 variants (Fig. 2d and Supplementary Fig. 4). These examples of suppression reconstitute multiple aspects of WT-like Inducibility traits, including low gene expression in glucose, high expression in galactose, and high growth rates in galactose and can be considered as examples of "harmonious" allelic combinations[15]: alternative genetic solutions to the core Inducible phenotype of the GAL pathway.

**Harmonious combinations can rewire the network**. We hypothesized that the harmonious combinations of GAL80 and GAL4 alleles could simply reflect reconstitution of the original regulatory network that exists in WT cells, with all GALR genes functioning as in the WT network. Alternatively, the functionality of these mutants might reflect different solutions to the same regulatory task, with the roles of other genes changed[16,17]. To distinguish between these two possibilities, we tested whether mutating the additional GALR genes, GAL3 and GAL1, had the same effect in these combinations of GAL80 and GAL4 alleles as in the WT system.

We tested a subset of GAL80 and GAL4 alleles in a combinatorially complete set of genotypes incorporating additional third- and fourth-order deletions in the potential galactose sensors GAL3 and GAL1 (Fig. 3 and Supplementary Figs. 6, 7, 8). All Inducible allelic combinations depended on GAL1 or GAL3 for robust growth, reflecting the importance of galactose sensing for pathway induction (Fig. 3b, c). However, the consequences of deleting GAL3 or GAL1 sensing activity varied extensively across the different genotypes. For example, whereas the WT switch is completely dependent upon GAL3 (Figs. 1d and 3), GAL3 was not required for induction in the Leaky GAL80.07 mutant (Fig. 3), nor was it required for Constitutive expression in GAL4C mutants (Fig. 3d). Moreover, GAL80S-1 + GAL4C double mutants were still Inducible when GAL3 was deleted (Figs. 3d, e and 4a, Supplementary Figs. 6 and 7, and Source Data).

GAL1's galactokinase activity is required for growth in galactose, so the dependency of a network on GAL1's sensing activity cannot be determined by simply deleting the gene. To test the dependency of each genotype on GAL1 sensing activity, we therefore used a strategy in which GALK genes from other species with different galactose sensing mechanisms were expressed from the GAL1 promoter ("GAL1::GALK", Fig. 3a, Methods). Consistent with GAL1 sensing activity being dispensable for induction in WT cells, replacing GAL1 with GALK from other species had no effect on gene expression or growth (Fig. 3b, c and Supplementary Figs. 6 and 7). In the Leaky GAL80.07 background, replacing GAL1 by GAL1::GALK also had no effect. However, in inducible GAL4 backgrounds, it completely prevented growth and expression when GAL3 was also deleted.

In contrast, in GAL4C mutants, GAL1::GALK reverted the Constitutive expression phenotype to a Leaky or Inducible phenotype (Figs. 3d and 4a). This would be consistent with the GAL pathway of GAL4C single-mutant variants being de-repressed in glucose not only because of reduced GAL80 repression (Fig. 3d), but also due to positive feedback via GAL1, which, through a baseline level of leaky expression (Fig. 3d) and ability to repress GAL80[9], could reach sufficient abundance to constitutively activate the system (Fig. 4a). Finally, in GAL4C + GAL80S-1 double mutants, GAL1::GALK reduced growth rate and expression more than deleting GAL3, including one combination where GAL1 sensing activity was essential (Figs. 3b–e and 4a).

Our dataset therefore contains multiple examples of "harmonious" combinations that exhibit the key WT-like Inducible phenotype of low expression in glucose, high expression in galactose, and high rates of growth in galactose (gray squares in Fig. 4a). Four are particularly notable in their altered dependence on the GAL3 and GAL1 sensor genes ("HC*" squares in Fig. 4a). The first combination is the WT itself, which depends on GAL3 completely but does not require GAL1 sensing activity in these conditions. The second combination is GAL80.07 + WT GAL4, where GAL3 is no longer essential and where deleting GAL1 has no effect except in combination with a GAL3 deletion. The third combination is the Inducible GAL4C + GAL1::GALK background, where GAL3 serves as a single essential sensor, and where GAL1 sensing activity is deleterious, causing high expression in glucose. Finally, the combination of GAL80S-1 + GAL4C has a flipped dependence on GAL3 and GAL1 compared to the WT system, with GAL3 no longer essential and growth and expression dependent on GAL1.

## Discussion

In summary, by combining mutations with diverse individual effects, our approach provides a more complete view of how variants in different genes combine to alter the activity of a model regulatory network. We found that the phenotypes of thousands of pairs of mutations fell into a small number of phenotypic classes, with the classes of individual mutants predicting very well their combined effects. In other words, once the phenotype of a single mutant is measured, its effect in combination with many other mutations is normally straightforward to predict, even when there is strong epistasis between loci. If this is also true for other systems including human disease, it will mean that accurate phenotypic prediction will often be possible without the need to resort to detailed mechanistic models, provided measurements of intermediate phenotypes such as gene expression can be made.

Rare combinations of mutations in the GALR genes did, however, have unexpected outcomes, including combinations of individually detrimental mutations that reconstitute a functional regulatory switch. These and other "harmonious" combinations of mutations constituted a functional system not because they resuscitated the original WT switch but because they formed alternative "re-wired" regulatory networks with altered "functions" for additional components (Fig. 4b). For example, combining the individually deleterious GAL4C with GAL80S-1 variants not only restored an Inducible system, an example of reciprocal sign epistasis, but also switched the system's dependence on the other two GALR genes, GAL3 and GAL1, an example of higher-order epistasis. Such changes in gene essentiality have been widely observed between and within species[18–22], but the genetic causes are poorly understood. The altered requirement for GAL1 and GAL3 across combinations of mutations in GAL4 and GAL80 shows that selective pressures on paralogous genes can substantially change with variation in other molecular players.

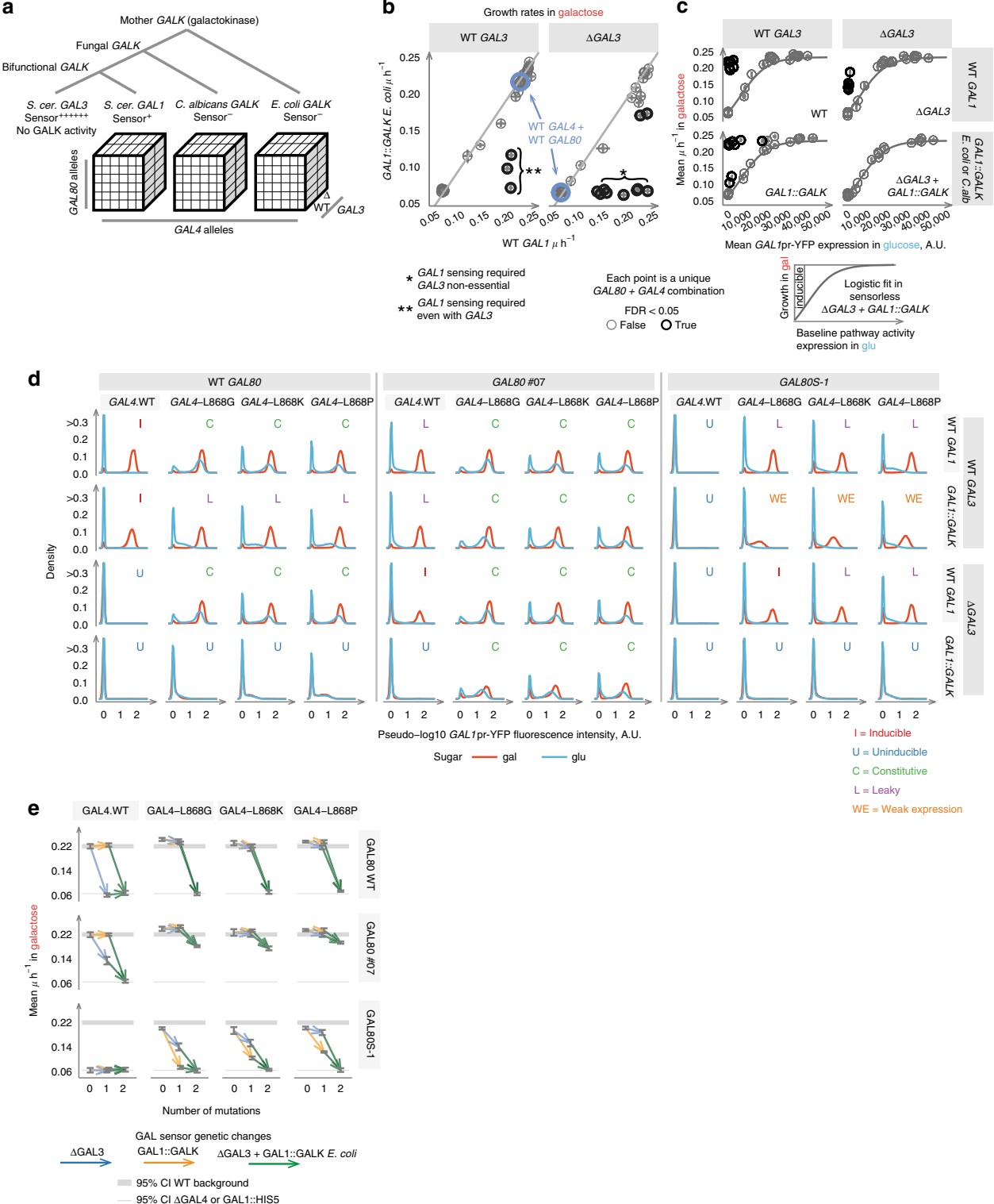

The molecular mechanism underlying the *GAL4C* + *GAL80S-1* double-mutant phenotype is intriguing. The *GAL* pathway is tightly repressed in glucose for *GAL4C* + *GAL80S-1* backgrounds, suggesting that *GAL80S-1* associates more tightly than WT *GAL80* to *GAL4C* variants. Similarly, the dominant-repressive nature of the *GAL80S-1* mutant is suppressed in the *GAL4C* backgrounds, allowing high expression in galactose. This implies that *GAL80S-1* must not only bind less efficiently to Gal4Cp, but

also retain the ability to interact with Gal1p and Gal3p to allow relief of repression in galactose media.

Mechanistically, it is still not clear why this mutant background exhibits a higher dependency on *GAL1* compared to *GAL3*. For example, we did not observe this in other *GAL80S* + *GAL4C* pairings. Most other *GAL80S* variants showed abilities to repress *GAL4C*, however tended to exhibit more "Constitutive" or "Uninducible" characters, with similar degrees of activation in

**Fig. 3** Higher-order genetic interactions in the GALR pathway. **a** Experimental design to determine the requirement for *GAL1* and *GAL3* sensor activity for *GAL* pathway induction across combinations of *GAL80* and *GAL4* alleles (Methods). **b** The effects of deleting *GAL1* and *GAL3* change depending on *GAL80-GAL4* combinations. The mean for the WT *GAL80-GAL4* pairs is indicated as a blue circle. Samples deviating significantly from the gray line where $Y = X$ are colored red (two-tailed *t*-test, FDR < 0.05 and mean effect ≥ 0.03 μ/h). These points reflect harmonious genetic combinations that depended on *GAL1* (left facet panel) or *GAL1* and *GAL3* (right facet panel) for high rates of growth. **c** Requirement for *GAL1* or *GAL3* for high growth rates from an initially uninduced state. In the *ΔGAL3* + *GAL1::GALK*-double mutants, the mean expression level for each *GAL4-GAL80* pair in glucose is used as a measure of *GAL* pathway "leakiness" or constitutivity to generate a null expectation for growth rate (black line, logistic fit). Samples colored red grew significantly faster (one-tailed *t*-test FDR < 0.05) than this expectation. Each point in **b** and **c** is the mean growth rate observed for a *GAL80-GAL4* pair in a given *GALK* and *GAL3* background ($N = 8$ across four independent transformations). **d** Distributions of *GAL1*pr-YFP expression for alleles across four genotypic dimensions. Lines and shading reflect within-genotype mean and SD. *GAL1::GALK* backgrounds bear the *E. coli GALK* construct. **e** Effects of *GAL1::GALK* and *ΔGAL3* on growth rate across *GAL4-GAL80* pairings. Lines originate and terminate at mean values for the given genotype and gray bars indicate the 95% confidence interval

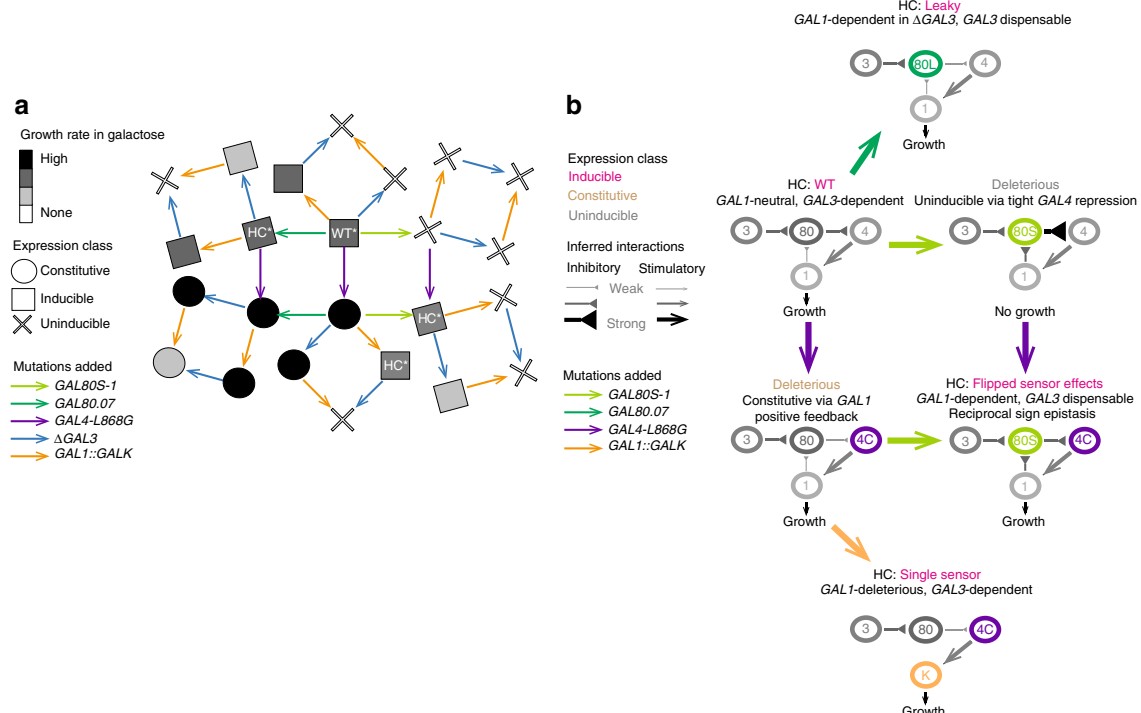

**Fig. 4** Rewiring of regulatory networks in harmonious genetic combinations. **a** Snowflake fitness landscape showing the dependency of *GAL4-GAL80* genotype pairings on *GAL1* and *GAL3* sensor activity. The shape of each node indicates expression phenotype and shading the growth rate in galactose. While many "harmonious" combinations of WT-like phenotypes are present, we highlight particular combinations to illustrate where galactose sensing requirements have changed appreciably (HC*). While all harmonious combinations require some combination of *GAL1* or *GAL3* for growth and expression in galactose, the dependence on *GAL3* and *GAL1* varies across every *GAL4-GAL80* background. **b** Rewiring of the GALR network across harmonious combinations. Each network's inferred wiring is illustrated for a given allelic combination, including Leaky *GAL80.07* ("80L"), super-repressor *GAL80S-1* ("80s"), the *GAL1* sensor deletion construct *GAL1::GALK* ("K"), and Constitutive *GAL4-L868G* ("4C"). "Inducible" classes include Leaky phenotypes, which show low but detectable expression levels in glucose and high expression in galactose

glucose and galactose environments that did not depend on *GAL1* or *GAL3* (Supplementary Figs. 6 and 7). This suggested that these mutants might bind less strongly to Gal1p and Gal3p in addition to (or instead of) tighter association to Gal4p. Mutant combinations including *GAL80S-2* in combination with *GAL4-L868K* or *GAL4-L868P* exhibit partial Inducibility in galactose, with highly leaky expression in glucose but increased expression and growth in galactose that is partially dependent on *GAL1* and *GAL3* (Supplementary Figs. 6 and S7). However, in contrast to the *GAL4C* + *GAL80S-1* backgrounds, in these cases the increased expression in galactose is more dependent on *GAL3* than on *GAL1* (Supplementary Figs. 6 and S7). Given these results, we conclude that the *GAL80S-1* + *GAL4C* backgrounds represent a special case in their increased dependency on *GAL1* compared to *GAL3*. It seems plausible that, compared to the WT Gal80p, the

Gal80S-1p variant exhibits altered binding affinity towards Gal1p in addition to Gal4p.

Taken together, our results illustrate how the genetic interactions between diverse alleles can be accurately predicted from gene expression in single mutants using models that capture the "stereotypical" epistasis in a system. However, they also demonstrate the importance of rare unexpected pairwise and higher-order epistasis for the fitness of individual genotypes, and illustrate how combinations of mutations can flip the essentiality of additional genes and rewire regulatory networks.

## Methods
**Yeast and *E. coli* strains**. Primers, strains, and specific construction notes for plasmids are in Supplementary Tables 1 and 2, respectively. All *S. cerevisiae* yeast strains were generated starting from BY4741 (MATa his3Δ1 leu2Δ0 met15Δ0

ura3Δ0[23]). All cloning in *E. coli* was performed using DH5-alpha or its commercial derivative NEB 10-Beta (New England Biolabs product number C3019I or C3020). Genomic DNA of BY4741, DH5-alpha and *C. albicans* strain SC5314 were used as templates to generate GALR and GALK constructs.

**Polymerase chain reaction**. For high-fidelity PCR reactions we followed manufacturer's instructions using either Extaq (TaKaRa # RR001C), KOD hotstart (Merck Millipore # 71086) or Q5 (NEB # M0491) with 0.6 μM final concentration primers.

For mutagenic PCR, 25 μl reactions of standard Taq (New England Biolabs # M0273) were used with plasmids bearing WT templates of the *GAL* genes (pAMN14 (*GAL3*), pAMN15 (*GAL80*) or pAMN31 (*GAL4*)). Initial template concentration was varied to allow 8–12 duplications of template.

Components added to mutagenic PCR (order: *Material, volume in μl*;): Standard Taq buffer (10×), 2.5; 50 mM MgCl2, 2.75; dCTP (100 mM), 0.25; dTTP (100 mM), 0.25; dATP (100 mM), 0.05; dGTP (100 mM), 0.05; MnCl2 (50 mM), 0.25; oligo1 (20 μM), 0.625; oligo2 (20 μM), 0.625; template (5 μl per 50 μl reaction), variable; Taq DNA pol. (5U/ul), 0.25; Water, 14.9.

**Generation of yeast chassis strains AN612 and AN634**. Primers and strains are in Supplementary Tables 1 and 2. AN612, a Gal1p-YFP fusion, was used for the first two experiments, and AN634, a *GAL1*pr-YFP transcriptional fusion was used for the final experiment. Standard lithium acetate transformation was used in all stages. BY4741 (S288c MATa ΔHIS3 ΔLEU2 ΔMET15 ΔURA3) was used as a starting strain. Starting from BY4741, we first integrated the YeCitrine-KANMX cassette from pKT140 as a fusion protein with *GAL1* protein (AN612) or as a disruption of the *GAL1* reading frame (AN634). KANR clones were screened at the cytometer, and those that exhibited high-FITC-A measurements in galactose and autofluorescent-equivalent levels of FITC-A signal in glucose were saved. For deletion of GALR genes, full loci, including promoter and terminator sequences were deleted (the deleted sequences was exactly the same as the sequence of the genes used as complementation cassettes in plasmid complementation constructs). Upon transformation, potential deletion transformant clones were screened by phenotype. For GAL3, clones with GAL80 + GAL4 WT backgrounds were screened by inoculating a 2-day-old colony (1 colony = $10^7$ cells) into 200 μl in 96-well plates containing YP + 2% galactose media and making four fourfold serial dilutions. After 12–18 h, those with no expression of the GAL1-YFP construct were selected and saved as the correct deletion strains. For ΔGAL80 strains in the GAL4. WT background, after integration of the deletion cassette, clones were screened for constitutive expression of GAL1-YFP. For deletion of GAL4, clones were screened for lack of GAL1-YFP expression similarly to the GAL3 clones. To make the combinations of allele deletions, clones were generated in the the appropriate order to allow phenotypic characterization. For ΔGAL3 GAL80.WT ΔGAL4 strains, after deletion of GAL3 locus, ΔGAL4 integrants were screened for integration of the deletion cassette at the GAL4 locus by traditional PCR screening of 5′ junctions. The final resulting strains, AN612 and AN634 were screened by PCR at the end for proper integrations of all constructs. To complement GALK activity in AN634, PCR products from plasmids pAMN50 (GAL1pr-GALK *E. coli*), pAMN51 (GAL1pr-GALK *C. albicans*), pAMN52 (GAL1pr-GAL1) and pAMN53 (GAL1pr-HIS5 *S. pombe*) were transformed into the GAL4::URA3 locus.

**Generation of screening plasmids for mutagenesis**. Primers and strains are in Supplementary Tables 1 and 2. Supplementary Tables 3–11 include information specific to the PCR products and transformations used. To ensure that all phenotypic variation observed in the PCR mutagenesis experiment were due to the locus targeted by mutagenesis and not the result of mutations in the other two GALR loci (potentially generated in the assembly or PCR process), we generated a screening plasmid with unique cutting sites between the GALR genes. These plasmids were generated by assembling three-GALR plasmids with the targeted GALR flanked by NotI sites and linker sequences. One plasmid was assembled per GALR locus. After in vivo gap-repair assembly of these plasmids, they were prepped and transformed into *E. coli* and screened for correct digestion patterns. These plasmids were named pAMN26, pAMN27, and pAMN28. After identifying correct banding patterns and phenotypic behavior, plasmids were prepared for downstream analysis by removal of the gene of interest. The locus of interest for each one (GAL4, GAL3, and GAL80, respectively) could be liberated with a NotI digest. Recircularization of these plasmids with a 5-min room-temperature T4 ligase reaction and transformation into *E. coli* yielded plasmids pAMN32, pAMN33 and pAMN34 with deletions of their locus of interest.

Each GALR-specific screening plasmid (pAMN32, pAMN33 and pAMN34) was digested at a concentration of 100 ng/μl overnight in 7 ml in a 15 ml Falcon tube at 37 °C with NEB Cutsmart Buffer + 10 units of NotI-HF (New England Biolabs # R3189) per ml. After at least 12 h' incubation, the reaction was brought to 30 °C, and ten units of Mungbean nuclease added to chew away overhanging ends. After 1 h incubation at 30 °C, reactions were terminated by extraction of enzyme and other protein with one volume of TE-buffered phenol/chloroform/isoamyl alcohol mixture at pH 8.0 (Sigma Aldrich P-2069). Extracted samples were washed twice with one volume of ether to remove excess phenol. Ether was dried off under a flow hood for 30 min. DNA was precipitated by addition of 1/10 volume of 3 M

sodium acetate and one volume of isopropanol, frozen at –80 for 20 min to overnight, and then centrifuged 20 min in a bucket centrifuge at 4 °C. Pelleted DNA was washed twice with 2 ml of room-temperature 70% EtOH (10 min spins each). EtOH was pipetted off and tubes allowed to dry. DNA was resuspended in 1 ml of TE buffer and purity and concentration checked on a Nanodrop and digest confirmed by running on an agarose gel. These vectors were used downstream as recipient vectors for mutagenic PCRs.

Primers used to generate mutagenic PCR fragments contained blunt-ended NotI scar sites + novel PmeI digest sites + linker sequences, allowing in vivo gap-repair assembly and downstream confirmation of plasmid structure (via PmeI dual cuts + a mid-vector NdeI site). Vectors for the single GALR mutant screen were co-transformed with mutagenized PCR product using high-efficiency transformation into yeast and clones selected and screened as described below.

**Generation of mutagenic libraries and targeted mutations**. Supplementary Tables 1, 2, and 3 describe strains and primers. See supplementary tables 4, 5, and 6 for information specific to the PCR products and transformations used. Primers for these constructs encode a linker, a short barcode, a NotI cloning site, and template binding sites for amplification. Targeted mutations and random mutants generated in this study used NotI-digested plasmids pAMN32, pAMN33, and pAMN34 to receive one or two fragments of the given GALR gene. Each plasmid generated in these steps were used downstream as templates in the combinatorial genetics experiments.

**Generation of combinatorial mutant constructs**. Supplementary Tables 1, 2, and 3 describe strains and primers. See Supplementary Tables 7 and 8 for information specific to the PCR products and transformations used. All assemblies for combinatorial genetics used four PCR products (the vector + one allele of each of the three-GALR loci) and were co-transformed into chassis strain AN612 or AN634. pRS415 (LEU2 marker) was used for combinatorial genetics experiments.

**Generation of GAL1pr-GALK constructs**. Supplementary Tables 1, 2, and 3 describe strains and primers. See Supplementary Tables 9, 10, and 11 for information specific to the PCR products and transformations used. Plasmid pAMN45 (pGAL1pr-MET15) was first generated, and GALK orthologs or HIS5 from *S. pombe* were cloned downstream of the GAL1 promoter. MET15 was used as a marker because all usual markers in the final chassis yeast strain AN634 were used. The final constructs were designed to disrupt the CaURA3 cassette that was used to delete the GAL4 locus. pAMN45 was made by six-fragment PCR product assembly and was used downstream as a template for parts to assemble the final GALK constructs. Clones bearing plasmids assembled in these transformations were screened for the ability to grow in 0.2% galactose and GAL-inducible growth in SC-HIS + 0.5, 0.1, 0.2, and 2% glucose ± 0.2% galactose media. Correct clones were used downstream to transform strain AN634 with a GAL4::URA3::GAL1pr-GALKxx-Met15 PCR product.

**High-efficiency gap-repair transformation in yeast**. In vivo homologous recombination ("gap repair") was used for generation of all plasmids except pAMN31, which was made by Gibson cloning. For gap-repair, yeast clones were generated using combined with a high-efficiency yeast transformation adapted from on a protocol described in ref. [24]. Appropriate yeast strain backgrounds were streaked from the freezer and incubated 2–3 days at 30 °C. A patch of cells were picked from the center of the streaked cells to prevent genetic bottlenecks of single clones. The cells were inoculated into 5–50 ml of YPD and grown 12–16 h until saturation with vigorous shaking at 30 °C. These overnight cultures were then inoculated into 30 °C YPD media to a 1 cm OD600 absorbance of 0.3, ~$2 × 10^7$ haploid BY4741 cells per ml. Cells were incubated in Pyrex glass bottles with vigorous shaking every 30 min to 1 h. After 4 h, cultures were aliquoted, appropriate to the scale, to 50 ml falcon tubes or 250 ml centrifuge bottles and centrifuged at 1250 x *g* for the the time required to pellet the cells (~2 min for 50 ml falcon tubes and 5 min for the 250 ml centrifuge tubes). Cells were resuspended in 10–50 ml of SLAT buffer, and centrifuged again in 50 ml falcon tubes for 5 min at 1250 x *g*, enough to pellet most cells.

Thereafter, per 50 ml of culture: cells were resuspended in SLAT buffer by agitating the cell pellet with a long pipette, followed by vigorous shaking. The volume of SLAT was adjusted depending on the volume of DNA transformation product being added such that a final volume of SLAT was 2.5 ml per 50 ml original culture. The protocol differed hereafter for library vs. targeted GALR allele clones.

For library generation, DNA was directly added to cells, including 50 μl of recently boiled-then-snap-chilled 10 mg/ml ssDNA (salmon sperm DNA Agilent # 201190), 1 μg vector, and 1 mg each PCR product for in vivo gap-repair assembly. The mutagenized PCR product was added directly to the cells. Cells + SLAT + DNA mixtures were left with occasional agitation at room temperature for at least 30 min. Thereafter, 10 ml of PLATE mixture was added to cells, and cell-SLAT-DNA-PLATE mixtures were shaken and left to rest at least 30 min at room temperature. Then dimethylsulfoxide (DMSO) was added to PLATE + cell mixtures to 8% final concentration, and they were heatshocked in a water bath for 20 min. Cells were then centrifuged 5 min at 1250 x *g*, resuspended in 0.5 M

sorbitol + YPD media and allowed to recover 1 h at 30 °C. After recovery, cells were centrifuged again at 1250 x g, and pellets resuspended in selection media and either recovered in 500 ml of liquid (200 RPM shaking for 48 h) or spread onto solid plates.

For high-throughput gap-repair assemblies with specific combinations of GALR alleles, cell mixtures were resuspended in 1.375 ml SLAT, 50 µl of recently boiled-then-snap-chilled 10 mg/ml ssDNA and 1 mg PCR vector per 50 ml original culture. Then 11 µl of cells + SLAT were aliquoted to 9 µl of DNA-SLAT mixtures pre-aliquoted to 96-well polyethylene PCR plates (Thermo Scientific AB-0700). Each gap-repair assembly mix had 0.1 µl of each of the three-GALR's PCR product + 8.7 µl SLAT. No shaking or agitation/mixing was used to blend the cells with DNA. After adding cells to 96-well plates, 80 µl of PLATE was added and plates sealed carefully using Biorad Microseal B seals. Samples were shaken vigorously by inversion to mix cells with the PLATE buffer. Cells were centrifuged in a swinging bucket centrifuge at 300 rpm for 3 s and then left at room temperature for at least 30 min and up to 3 h. After incubation, plate seals were removed and 8 µl of DMSO was added to cell + PLATE mixtures. Seals were reapplied to the 96-well plates and the samples shaken vigorously by inversion, followed by centrifugation in a swinging bucket centrifuge at 300 rpm for 3 s (to bring samples down to the tube bottoms), followed by heat shock at 42 °C for 20 min in a pre-warmed PCR block.

After heat shock, cells were centrifuged for 1 min at 1250 x g, PLATE + SLAT mixture dumped out, and cells resuspended in 80 µl YPD + 0.5 M sorbitol and incubated for 1 h. Plates were then spun down for 3 min at 1205 x g to pellet cells, and cells were resuspended in 500 µl of SC-LEU + 2% glucose + ampicillin 100 mg/l selection medium in 96-well plates. Plates were sealed with Biorad Microseal B seals or unsealed and covered as a stack of plates with the same plastic sheaths in which the plates were shipped. Efficiency of transformation of each well could be assessed after 36–48 h growth by counting colonies on the bottom of the wells. Transformations with fewer than 20 colonies could be accurately counted. Although most wells had >20 colonies, those with <10 single colonies were excluded from analysis downstream. Cells were then resuspended and 3 µl inoculated into 75 µl of selection media and allowed to grow overnight. Cells were resuspended with 75 µl of 50% glycerol and frozen at −80 °C until experiment measurement.

**Pre-growth of clones before flow cytometry.** To begin flow cytometry analysis, plates with combinatorial genetic assemblies were taken from freezer and in total sat at room temperature for 20–25 min. Plates were put on orbital shakers once thawed and shaken 1–5 min before inoculation of 10 µl into 190 µl (200 µl final volume) of SC-[LEU or HIS] + 0.1% glucose + 100 mg/l ampicillin a + 20 mg/l chloramphenicol, and grown without agitation in stacks encapsulated in the plastic sheaths in which the plates were shipped (Sarstedt 82.1581) for 12–24 h to saturation. Cells were resuspended on the plate shaker and diluted 1/50 into 75 µl SC-[LEU or HIS] + 0.1% glucose (no antibiotics here) grown 18–24 h to saturation in stacks of unsealed plates encapsulated in the plastic sheaths in which the plates were shipped in preparation for inoculation to galactose and measurement at the cytometer the next day.

**Growth and YFP measurement of cells by flow cytometry.** Plates containing 75 µl 0.1% glucose-grown cultures (either from the single clones picked in the mutagenesis experiment or the clones generated to have targeted allele combinations) were grown 18–24 h unsealed in stacks of plates encapsulated in the plastic sheaths in which the plates were shipped (Sarstedt 82.1581). After growth, samples were placed on an orbital shaker for 1–5 min to resuspend cells, then 150 µl of ddH2O was added to the cells to make a threefold dilution of the original cell density, with continued shaking for another 1–5 min. Nine microliters of the ddH2O-diluted cultures was added to 4 °C plates containing 141 µl of 1.06x concentrated SC-[LEU or HIS] + 0.2% galactose media. Inoculated galactose plates were sealed with Microseal B seals and placed immediately at 4 °C to prevent growth or gene expression prior to beginning of growth experiment. We found that only turbid cultures could be resuspended by shaking on the 2.5-mm-radius orbital shaker. Therefore, while inoculating into galactose, we took care to distribute the cells evenly across the whole well. Plates were then sealed with Microseal B seals and put at 4 °C. At the end of the day all glucose-pregrown cultures that had been inoculated into galactose media were placed at 30 °C in stacks of 1–2 plates to begin growth.

After inoculation of galactose plates with glucose-grown cells, we put the glucose plates at 4 °C until measurement at the flow cytometer (BD FACS Canto; FACS Diva v 5.0.3 Firmware V 1.4). Prior to measurement at the cytometer, plates were put back on the shaker for 2 h. Plates were visually inspected to be sure that the cells in all wells were well-suspended. We measured cell density and gene expression (bandpass filters "FITC-A" 530 ± 15 nm and "PE" 585 ± 21 nm were used for YFP signal and 488 ± 5 nm for SSC signal). High-throughput sampling mode was used with no mixing. The median time to complete a plate was 18 min. During this time, we determined that cell density measurements did not appreciably change. If any problem was encountered during the cytometry and the measurements needed to be stopped, we took the plate out and put back on the plate shaker briefly to resuspend the cells before resuming the cytometry.

After 12 h of growth at 30 °C in SC-[LEU or HIS] + 0.2% galactose, samples were placed on ice or on a cold surface in a 4 °C room to arrest growth and allowed

to cool at least 30 min prior to exposure back at room temperature. Prior to measurement at the cytometer, Microseal B covers were removed and samples put on the orbital shaker for 2 h covered by a breathable plate seal. As mentioned above, samples that did not grow appreciably could not be easily resuspended by the orbital shaker. Therefore, prior to measurement, all cultures were pipetted up and down five times with a multichannel micropipette, and then placed immediately in the FACSCanto for analysis. Prior to sampling in the cytometer, wells were scored by eye for high growth or low growth. The cytometer template's sampling rates were adjusted according to these by-eye scores: high-density samples were sampled at 0.5 µl per second, while low-density cultures were sampled at 2.0–3.0 µl per second, with occasional intermediate sampling rates for obviously intermediate cell densities. Each sample's sampling rate can be found in all supplementary tables where we report data for these experiments.

**Selection of single clones for combinatorial genetics.** After screening and phenotypically characterizing mutagenized GALR variants by flow cytometry (see above) we isolated plasmids of interesting phenotypes. Clones were selected to reflect either outlying phenotypes or more typical behavior. "Outlying phenotypes" included samples where both fracon.glu and fracon.gal measurements were >0 and <1, indicating that the clones had a constitutive character but could not fully induce the GAL pathway. Another rare phenotype we tried to isolate were clones where mean signal in ON cells was less than mean signal of typical inducible ON clones (as discussed in the text this phenotype was quite rare). Although some GAL3 mutants appeared to have constitutive characters in the first screen, we found that none of these phenotypes were recapitulated upon subcloning.

After selection based on phenotype, clones were thawed from the freezer and struck to single colonies. These were inoculated into SC-HIS + 2%glucose media in PCR plates and genomic DNA (gDNA) prepped in 96-well plates as described above. Clones were transformed into electrocompetent 10-Beta cells (New England Biolabs # C3020) using a 96-well plate electroporator using fresh electroporation plates (BTX 45–0450-M). Cells were recovered in deep-well plates (Thermo Scientific # 260252) and recovered in 0.6 ml SOC media for 1 h prior to inoculation into 0.6 ml LB + 100 mg/ml ampicillin. The next day plasmids were prepped in 96-well plates as described above. Preps were digested with NdeI and PmeI enzymes, which yielded three bands in correctly assembled constructs. Plasmids with the correct banding patterns were used downstream in the combinatorial genetics experiment.

**Analysis of the raw flow cytometry data.** R was used for all analyses. Scripts and data are available at the github link: https://github.com/AaronMNew/HarmoniousCombinations. FCS3 files were exported from the computer controlling the FACSCanto measurements and sampling rate information extracted from exported .xml files generated from export of "Experiment Template". Scripts for extracting metadata from these .xml files are found in the supplementary code. Experiment "layout" files were generated including clone information, known genotype information and censorship information (censored either if they had very low transformation efficiency or a contamination), and this was merged with metadata of sampling rates in the .xml file. As a basic overview of the analysis, the *Bioconductor FlowCore* package tools were used to open the binary FCS files and filter first based on cell shape and size information using first a rectangle including 95% of observations in side scatter (SSC) and forward scatter (FSC), then a centroid algorithm was used to identify the most dense observations in these two dimensions, excluding between 30–50% of outlying original observations. FITC-A signal was used to quantify YFP expression. The predicted FITC-A value of a PE-A reading was predicted by a linear model of FITC-A signal as a function of the correlated signal PE-A using the function *lm (log(FITC-A)~log(PE))*, and these predicted values were used in the rare cases where a cell's FITC-A signal exceeded the machine's maximal measurement value. Then, key parameters of FITC-A distributions were extracted, including the mean YFP signal, fraction ON (the proportion of cells falling above an empirically determined cutoff based on auto-fluorescent cell controls). As many lowly expressing cells gave negative values at the flow cytometer, we calculated a pseudo-log10 FITC-A measurement as the log10 value of of the raw FITC-A plus 1000 A.U. –3 log10 arbitrary units using the function $log10(raw\ FITC\text{-}A\ measurement + 1000) - 3$. These pseudo-log10 fluorescence intensity values were broken into 60 equally spaced bins using the function *cut()*. Cells were counted in the glucose and galactose environments by determining the slope term corresponding to events/second information of each FCS file using a linear model *lm (events~ms)*. Glucose-grown biological replicate clones were then matched with their next-day galactose measurements. We calculated cell densities by multiplying the events/second measurement by the known sampling rates extracted from the .xml file. The density of the culture in galactose was calculated as this measured cell density parameter multiplied by the inverse of the dilution factor from glucose the day before (150/9). The log2 change of the culture $Log2(Density.GAL.grown.culture/Density.GLU.grown.Culture)$ was scored as the number of generations, and the growth rate parameter $\mu$ was calculated as the Malthusian growth rate parameter determined by the natural log of of the final density of the culture minus the natural log of the initial density divided by the number of hours of growth $(log(Density.GAL.grown.culture/Density.GLU.grown.Culture)/12\ h.)$

**Plotting and basic analysis of data**. All scripts are online at github https://github.com/lehner-lab/HarmoniousCombinations. Custom functions, *data.table* and *ggplot2* packages were used for summarizing data and plotting. Minor esthetic changes were made in Adobe *Illustrator*. The final two panels of Fig. 4 were made in Adobe *Illustrator*.

**Gene expression distribution clustering**. For gene expression distribution clustering, we used the HDBScan* function (R package *dbscan*)[25]. This algorithm establishes hierarchical clusters based upon distance-weighted graphs assembled according to the density of data points across *n*-dimensions. Its final cluster assignments are mainly sensitive to the "minPts" parameter, which sets the minimum cluster size. This sensitivity arises primarily from a (1) size of the dataset (because more points for a large dataset will yield the same number of clusters as for a small dataset) and (2) how distantly the clusters of data points are spread in *n*-dimensional space.

For gene expression distribution clustering, we wanted to cluster based on expression in both glucose and galactose. For this, we first paired the densities of expression across 60 bins of pseudo-log10-transformed A.U. FITC-A signal for each sample at time 0 (glucose expression) and after 12 h of growth in galactose (galactose expression) to generate vectors of 120 units for each observation. The mean vector for each unique genotype was then calculated. These mean values were then pseudo-log transformed to exaggerate signal within bins exhibiting low-density values, for example such that a small fraction of cells active in glucose would be more salient to the HDBScan* algorithm (Supplementary Fig. 3A). For this, the density values for each bin transformed as the log of the density value rounded to the nearest 1/1000 plus 1/1000 as follows:

$$V = \log 10(\text{round}(v, 3) + 0.001)$$

Where $V$ is the final pseudo-log density value and $v$ is the original density value. A matrix was made comprised of rows corresponding to each genotype's vector of measurements of $V$, with pseudo-log10 A.U. expression distribution bins as columns. As a final step, each bin's $V$ across genotypes was scaled by z-score. The scaled 120 dimension matrix was then clustered using HDBScan* algorithm.

We evaluated clusters identified by the HDBScan* algorithm by how well the cluster assignments could explain total phenotypic variance, determined by a linear model of phenotypic values scaled by phenotype using the function

$$\text{lm (within\_genotype\_mean\_value} \sim \text{cluster} * \text{phenotype\_id)}$$

Where phentoype_id was one of the five phenotypes, including fraction ON in galactose, fraction ON in glucose, mean YFP expression in galactose, mean YFP expression in glucose, and the growth rate parameter $\mu$.

For experiment 1, with three clearly defined clusters, three clusters emerged consistently across minPts parameters. For experiment 2, across minPts parameters, two levels of clusters were evident: the broadest category included the three main inducible, uninducible, and constitutive clusters, and explained 84% of total phenotypic variance across all phenotypes. A more narrow set of classes comprised nine categories, explaining 97% of total phenotypic variance. Clusters were then manually curated to five intermediate "constitutive", "uninducible", "inducible", "leaky", and "weak expression" categories, which together explained 96% of total phenotypic variance. We chose the intermediate classifications for discussion in the main text for prediction of double-mutant phenotypes from single-mutant phenotypes due to their power to explain total phenotypic variance in the dataset compared to the broad categories and limited number of parameters compared to the more narrowly defined categories (see Supplemental Analysis Code).

This clustering was sufficient to cluster all single-mutant profiles. However, a certain 3–4% of double-mutant samples remained unclassified. A closer look at the these samples showed that most expression distributions were "flavors" of the other single-mutant phenotypes, however due to slight differences and their infrequent numbers, they remained unclustered. For example, the three GAL4 clones in the "weak expression" category all showed low-max gene expression level, however otherwise behaved WT, for example exhibiting constitutive characters in combination with ΔGAL80. Repeating the clustering with unclustered samples plus these less frequently observed weak expression clones showed that most unclustered samples showed characteristics similar to these GAL4 alleles, and indeed that they mostly included one of these three GAL4 single-mutant backgrounds. Using a similar approach, we took the remaining unclustered samples and repeated their clustering alone with samples from the three archetypal samples identified in the first experiment, leaving all previously uncategorized expression profiles with a classification. Finally, visual inspection revealed that some clones with constitutive characters exhibited a nonetheless high degree of activation in galactose relative to glucose. To quantify this we took the mean YFP signal in galactose/mean YFP signal in glucose as a measure of a clone's induction. Using this parameter, we could identify clones mis-classified as constitutive to be "inducible" or "leaky". Similarly, some "inducible" clones displayed low mean induction values, so were classified as uninducible, and some "uninducible" clones actually showed > 40% of cells ON in galactose with high a mode of expression and so were classified as inducible.

**Multiplicative model to predict growth rates in double mutants**. For calculating expected growth rate based on single-mutant effects, we used a first-order geometric model of phenotypic variation where the expected multiple mutant locus ($\mu_{MUT}$) phenotype is the product of the phenotypes of each (single) allele across loci in the WT background normalized by the WT reference phenotype value. So, for two loci:

$$\mu_{MUT} = \mu_{mut1} * \mu_{mut2} / \mu_{WT}$$

Standard errors for these predictions were propagated as $se_{mut} = abs(\mu_{mut}) * \text{sqrt}(((N-1)*(se_{WT}/\mu_{WT}) + se_{mut1}/\mu_{mut1} + se_{mut2}/\mu_{mut1})^2)$, where $N$ is the number of replicates. Phenotype values used for $\mu$ were equal to growth rate minus the background growth rate observed in the absence of any GAL regulator. These expectation values were used to predict overall variance in the dataset, explaining 52%, 20%, and 85% of variance for pairings between GAL3 vs. GAL4, GAL80 vs. GAL3, and GAL4 vs. GAL80, respectively. Overall this model explained 55% of variance across the dataset. These low values stem from the fact that pathway-level epistasis leads to constitutive and leaky expression profiles, which dominate signal in the dataset (Supplementary Fig. 5A). For example, all pairwise combinations of constitutive GAL80 and uninducible GAL3 are constitutive and therefore grow at high rates, when the prediction prediction is that they will grow slowly due to GAL3's low growth rate. Similarly, fast-growing constitive GAL4 variants are predicted to grow slowly in ΔGAL3 backgrounds, when the double mutant remains constitutive. Harmonious combinations of uninducible GAL3 paired with leaky GAL80 lead to leaky or inducible double-mutant phenotypes, which grow quickly. Similarly, harmonious combinations of GAL80S-1 and GAL4C permit high rates of growth, when they are predicted to grow slowly due to the dominant-repressive single GAL80S-1 backgrounds.

**Expression class prediction of growth rate in double mutants**. The expression clusters of each single mutant were matched to each double mutant. Mean growth rate values for each unique combination of single-mutant expression clusters were then taken as an expectation of the double-mutant's growth rate. For the 5-member Intermediate classification scheme, these 23 unique numbers were used to calculate the fraction of variance explained across all individual observations, and within-genotype mean growth rate values were calculated and plotted against predictions for the second figure. Note that these 23 unique combinations do not include all 32 possible combinations of $4 \times 4 \times 2$ GAL4, GAL80, and GAL3 classes. This is because certain triple-mutant classes were not measured. Specifically Uninducible GAL3 was never paired with the three classes of GAL80 and GAL4 that do not include Inducible. These classes number $3 \times 3 \times 1$ GAL4, GAL80, and GAL3 alleles = 9, so 32−9 = 23 unique clusters arising from single-mutant combinations.

Notably, these missing classes are all almost included in the final experiment, including constitutive GAL80, Leaky GAL80, Uninducible GAL80, Uninducible GAL4, Constitutive GAL4. The only one not included are the Weak Expression class of GAL4, three alleles of which all behaved very similarly to WT GAL4 (e.g., uninducible with uninducible GAL80S variants or uninducible GAL3 variants, and constitutive with constitutive GAL80 variants) but with lower peak expression level.

Next, we repeated the same analysis using the Broad classification (with three possible classes—Inducible, Uninducible, and Constitutive). With $3 \times 3 \times 2$ GAL4, GAL80, and GAL3 classes there were 18 possible combinations, but with four triple-mutant combinations not observed, there were a total of 14 possible pairwise combinations. Similarly, we examined model performance using the Narrow classification scheme, which consisted of eight total classes and 66 unique pairwise combinations.

We next performed a cross-validation by heavily downsampling our datasets based on class membership. For this, we calculated growth rate expectations based on sampling 1, 2, or 3 randomly selected alleles from each class, and predicted phenotypes for the remaining >4500 double mutants. We performed 1000 iterations of this approach for the Broad and Intermediate classification schemes. Median variance explained for these downsampled datasets ranged from 86% (one allele in the Broad classification parameter scheme, with 0.4% of the dataset downsampled for training to predict the remaining 99.6% of double mutants) to 92% (three alleles selected for the five-class Intermediate parameter scheme, with 6% downsampled to predict the remaining 94% of double mutants).

Finally, to test whether the locus identity was important for these models' performance, we blinded the models to the identity of the GALR locus. Thus, for the Broad three-class classification scheme, for any given double mutant, we generated an expectation of its growth rate based upon the mean growth rate of all other double mutants with matching underlying single-mutant gene expression classifications combinations, to generate six parameters (Inducible + I, I + Constitutive, I + Uninducible, C + C, C + U, and U + U). For example, to predict a double mutant whose single mutants were a Constitutive GAL80 variant paired with an Uninducible GAL3 variant (C + U), we computed the mean growth rate of all double mutants comprised of a Constitutive single mutant and an Uninducible single mutant. This mean value then included genotypes with a Constitutive GAL80 and Uninducible GAL3, a Constitutive GAL4 + Uninducible GAL3, a Constitutive GAL4 + Uninducible GAL80, or an Uninducible GAL4 + Constitutive GAL80. This analysis was performed with the Broad and Intermediate classifications, with six and 13 parameters calculated for each, respectively. This model was highly predictive, explaining 77% of genetic variance for the Broad

three-class scheme and 79% for the five-class Intermediate classification scheme. However, gross errors were made (Supplementary Fig. 5E). For example, for the Broad classification, all five combinations except C + U were predicted accurately regardless which GALR locus contributed the underlying phenotype. For example, an U + U all yielded low growth rates regardless which locus contributed the Uninducible single mutant. However, the model made gross errors in the cases of C + U because the molecular mechanisms for these phenotypes depended on the function of the underlying gene pairs involved. For example, all combinations of Constitutive GAL4 or GAL80 + Uninducible GAL3 were fast-growing Constitutive mutants, reflecting these variants' circumvention of the need for a functioning GAL3 gene. However, another C + U combination—Constitutive GAL80 + Uninducible GAL4 was slowly growing because presumably all GAL4 variants were likely detrimental to pathway transcription. Finally, the last C + U genotype were Constitutive GAL4 + Uninducible GAL80, and these double-mutant combinations exhibited highly variable phenotypes (see Fig. 2d and Supplementary Fig. 4), including re-wired WT-like double mutants.

**Logistic fit of growth rate to GAL1-YFP expression**. To demonstrate that GALK orthologs from E. coli and C. albicans did not display signaling activity, we compared expression of the GAL1pr-YFP fusion to growth rates in which all sensors were deleted and found a sigmoidal relationship between mean YFP expression in glucose. We used this latter expression in glucose a measure of "pathway leakiness" or constitutivity to predict the expected growth rate of the culture in galactose. For this, we fitted a logistic curve using R's SSlogis() function of growth rate as a function of initial gene expression level to demonstrate that inducible or leaky "harmonious combination" mutant backgrounds are able to mount an induction response and high growth rate from an initially OFF state. A one-sided t-test was performed for each within-genotype mean across all backgrounds compared to this null expectation. To account for the differential growth rates observed for C. albicans vs. E. coli GALKs, we fitted separate curves for calculation of t-test statistics, while the main figure simply shows the logistic curve across all E. coli and C. albicans backgrounds. See supplemental code for details.

**Standard lithium acetate yeast transformation**. PLI (Polyethylene glycol 3530 + LIthium acetate + 1x TE buffer) is 50% polyethylene glycol + 0.1 M LiAc in TE Buffer and was prepared ahead. Cells were inoculated 200 μl into 25 ml YPD and grown in Falcon tubes with occasional shaking at 30 °C for 4–5 h. Cells were pelleted by centrifugation on high in the Falcon tube in which they were grown, washed once with 700 μl 0.1 M LiAc, spun again quickly in the Eppendorf tube to pellet cells, and resuspended 200 μl of 0.1 M LiAc. 10 μl of boiled ssDNA (1 mg/ml) + 25 μl of PCR product was added and samples mixed by flicking. Tubes were optionally incubated at room temperature for up to 30 min. Six-hundred microliters of PLI was then added to cells and mixed well by vortexing. Tubes were optionally incubated at room temperature for up to 30 min. Tubes were then incubated at 42 °C for 30 min. Cells were pelleted at 1250 x g for 3–5 min and then resuspended either directly in selection media (for auxotrophic markers) or in 0.5–5 ml YPD media (for dominant drug resistance markers). YPD samples were incubated 3–4 h at 30 °C. Samples were then spread on solid selective media (1–5 plates depending on expected transformation efficiency). Clones were visible after 2 days and were struck to single colonies on selective media and grown for 48 h more. Clones that grew in the patches were picked for downstream phenotypic screening and genotypically and correct clones were frozen from single colonies.

**Small-scale genomic DNA (gDNA) prep**. Yeast genomic DNA was isolated by alkaline lysis and isopropanol precipitation using a scaled-down protocol based on that provided by MasterPure™ Yeast DNA Purification kit (#MPY80200). One-hundred fifty microliters of yeast cells were grown overnight in appropriate selective media in 96-well PCR plates covered with breathable seals with no shaking. The next day they were spun for 1 min in a swinging centrifuge centrifuge on high (~3200 x g or 4000 RPM on a swinging bucket centrifuge). Spent media was shaken out and immediately after dumping spent media, plates were swabbed on ethanol-soaked paper towel to remove most of the media still clinging to the sides of the plate. Cells were resuspended in 50 μl lysis buffer by pipetting or inversion while covered with Biorad Microseal B seals (catalog number fMSB1001) and inverted several times to mix. Plates were spun briefly for 1 s at 300 RPM to get the lysed cells back into the wells and off of the plate sealer. Samples were incubated at 65 °C for 15 min for lysis, and then plates placed on ice for 5 min. Plate seals were removed and 25 μl of MPC precipitation buffer was added to the lysed cells. Plates were re-sealed and inverted multiple times to be sure protein and other cellular debris was precipitated. Debris was pelleted by centrifugation on high (~3200 x g, 4000 RPM) in a bucket centrifuge for 10 min. Then, plate seals were removed and 50 μl of gDNA-containing super-natant was transferred into 50 μl of isopropanol in a new 96-well plate. Plates were re-sealed and inverted several times to mix the DNA and isopropanol. Samples were then centrifuged on high (~3200 x g, 4000 RPM) in a bucket centrifuge for 10 min. After centrifugation, isopropanol was dumped off and while still upside down, the plates gently dabbed on paper towels to absorb more isopropanol clinging to the plate. Sixty microliters of 70% ethanol was then added, plates re-sealed and inverted gently one time to mix the remaining isopropanol and ethanol together. Plates were then centrifuged on high (~3200 x g, 4000 RPM) in a bucket centrifuge for 2–5 min. An

optional second 70% ethanol wash was was sometimes performed. Then ethanol was dumped and while still upside down put the plate on a paper towel to absorb ethanol clinging to the plate. Plates were then spun briefly to bring remaining ethanol to the bottom of the wells, and using a Rainin multichannel P10 with LTS tips (very fine tips) the remaining ~4–10 μl of ethanol was removed from the plates. Plates were allowed to dry 10 min. DNA was resuspended in 25–50 μl EB buffer, mq H2O or TE buffer (depending on downstream use). Quality of prep was confirmed by measuring purity and estimation of DNA concentration on a Nanodrop and running an agarose gel of 3 μl of 12 preps randomly selected across the plate.

**Small-scale plasmid mini-preps**. To isolate plasmids, we used a simple alkaline lysis miniprep protocol using buffers P1, P2, and P3 from Qiagen (catalog numbers 19051,19052, and 19053 respectively), either at a "normal" scale, with 1.5 ml of saturated bacterial culture yielding > 10 mg of DNA or scaled-down in a 96-well plate with yields of >1 mg plasmid DNA per sample.

For normal mini-preps we first picked single colonies from a selective plate or inoculated a stab of cells directly from the –80 freezer stock into at least 2 ml of liquid LB containing the selective antibiotic and incubated overnight with vigorous shaking. After incubation, 1.5 ml of cells were pelleted at 13,000 rpm for 1 min. Cells were resuspended in 150 μl buffer P1 + RNAse, lysed for 1–5 min in 150 μl buffer P2, and then cellular debris precipitated with 150 μl buffer P3. Debris was then pelleted at 13,000 RPM for 10 min. Plasmid DNA was precipitated by adding one volume of isopropanol, the tubes inverted a few times to mix well, and then tubes were centrifuged at 13,000 RPM for 10 min at 4 °C. Isopropanol was dumped from the tubes and 0.5 ml of room-temperature 70% ethanol was added, the tubes gently inverted one time, and spun at 13,000 RPM for 2–5 min. Sometimes the pellets were washed again with ethanol. Ethanol was dumped, residual ethanol removed by a quick spin and pipetting, and the tubes left to dry for 15 min. DNA was resuspended in 1x TE buffer or EB buffer from Qiagen (catalog number 19086).

For 96-well plate mini-preps, a scaled-down version of the protocol used above was followed. Single colonies or pools of transformants from yeast clones were inoculated into LB + selection in a deep-well 96-well plate (Thermo Scientific 260252) and the plates sealed using breathable plate seals (Thermo Scientific # AB-0718) and incubated overnight with vigorous shaking on a 2.5 mm radius orbital plate shaker overnight at 37 °C. After growth, 150–160 μl of turbid cells were pipetted into 96-well PCR plate and centrifuged on high (~3200 x g or 4000 RPM on a swinging bucket centrifuge) for 5 min. Spent media was shaken out into an autoclavable or bleachable container, and immediately after dumping spent media, plates were dabbed onto an ethanol-soaked paper towel to remove most of the media still clinging to the sides of the plate. Cells were resuspended in 25 μl P1 buffer + RNAse by pipetting up and down, then 25 μl buffer P2 was added to cells. Plates were sealed with Microsea B seals and inverted several times to mix, then left to let sit for 5 min. Plates were spun briefly for 1 s at 300 RPM to get the lysed cells back into the wells and off of the plate sealer. Plate seals were removed and 25 μl of buffer P3 was added to the lysed cells and plates were re-sealed and inverted multiple times to be sure protein and other cellular debris was precipitated.

Debris was pelleted by centrifugation on high (~3200 x g, 4000 RPM) in a bucket centrifuge for 10 min. Then plate seals were removed and 60 μl of plasmid-containing supernatant was transferred into 60 μl of isopropanol in a new 96-well plate. Plates were re-sealed and inverted several times to mix the DNA and isopropanol. Samples were then centrifuged on high (~3200 x g, 4000 RPM) in a bucket centrifuge for 10 min. After centrifugation, isopropanol was dumped off and while still upside down, the plates dabbed paper towels to absorb more isopropanol clinging to the plate. Sixty microliters of 70% ethanol was then added, plates re-sealed, and inverted gently one time to mix the remaining isopropanol and ethanol together. Plates were then centrifuged on high (~3200 x g, 4000 RPM) in a bucket centrifuge for 5 min. An optional second wash was sometimes performed. Then ethanol was dumped and while still upside down put the plate on a paper towel to absorb yet more ethanol clinging to the plate. Plates were then spun briefly to bring remaining ethanol to the bottom of the wells, and using a Rainin multichannel P10 with LTS tips (catalog number 17005873; very fine and flexible tips) the remaining ethanol was removed from the plates. Plates were allowed to dry 10 min. DNA was resuspended in 25–50 μl EB buffer, mq H2O, or TE buffer (depending on downstream use). Quality of prep was confirmed by measuring purity and DNA concentration at a Nanodrop and running an appropriate digest on an 0.8% agarose TAE-buffered gel.

**Reporting summary**. Further information on research design is available in the Nature Research Reporting Summary linked to this article.

## Data availability
The source data underlying Figures and Supplementary Figures are provided as a Source Data file. Data used to generate the analyses and figures presented in this document can be found at [https://github.com/lehner-lab/HarmoniousCombinations]

## Code availability
All code used to generate the analyses and figures presented in this document can be found at [https://github.com/lehner-lab/HarmoniousCombinations]

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

## Acknowledgements

We thank members of the Lehner lab and L. Carey for comments on the manuscript, and O. Fornas, E. Ramirez, E. Julia, and A. Bote of the UPF/CRG Flow Cytometry Facility for flow cytometry expertize and assistance. This work was supported by a European Research Council (ERC) Consolidator grant (616434), the Spanish Ministry of Economy and Competitiveness (BFU2017–89488-P and SEV-2012-0208), the AXA Research Fund, the Bettencourt Schueller Foundation, Agencia de Gestio d'Ajuts Universitaris i de Recerca (AGAUR, 2017 SGR 1322.), and the CERCA Program/ Generalitat de Catalunya. A.M. New was supported in part by fellowships from EMBO ALTF 505–2014 and the Spanish Ministry of Economy and Competitiveness Juan de la Cierva fund.

## Author contributions

A.M.N. conceived the study, conducted experiments, analyzed data, and wrote the manuscript. B.L. conceived the study and wrote the manuscript.
