## [Peer Review File · Nature Communications]

Reviewers' Comments:

Reviewer #1:

Remarks to the Author:

In this paper, New and Lehner thoroughly dissect a model gene regulatory network by a combination of first, second, and higher order genetic perturbations. They evaluate a galactose reporter and cell growth rate in a large number of genetic backgrounds, and build models that allow predicting one from the other with good accuracy. The data and code supporting the publication are freely available. Overall, this nicely designed and reported study is an important landmark in and exemplar of the genetic variety that can underlie nominally wild type phenotypes, and demonstrates the relevance of higher order interactions.

Major comments

* Terminology. "Harmonious" combinations are also known as genetic suppression. Given the amount of confusion around terminology in discussing genetic interactions, perhaps it is appropriate not to introduce new definitions, and use definitions of masking, suppression, bypass suppression, etc. to describe the observations? I am not against the use of the word *per se*, but to the lack of clear link to the commonly used terms.

* Mechanisms of achieving harmonious combinations of mutations. Are the harmonious mutations in the repressor Gal80 and the activator Gal4 compensatory, individually destroying, but in combination restoring a physical interaction?

* Definition of harmonious. Figure 4A depicts a range of genotype-phenotype combinations, but only some are defined as harmonious, even though the phenotype is similar to that of wild type. Specifically, why are WT+Gal1:GALK and Gal80.07 + GAL1:GALK not considered harmonious?

* Phenotypic prediction. Only a few combinations of prediction are considered (growth from single mutant YFP values). It would be interesting [but not required] to also see how well single mutants predict double and higher order mutants, using both growth and YFP as predictors and targets in the four possible combinations. (Page 5 3rd paragraph). If the Authors choose not to, please explain why only a single direction of prediction is interesting.

Minor comments

- It would be nice to have the "leaky" and "weak" classes also annotated on Figure 2C. Otherwise, not clear e.g. where points at (0.25,0.5), (0.55,0.45), (0.65,0.75) are.
- The cluster centroids are off-diagonal in figure 2B. Figure S1B (expt 2 vs expt 3) suggests that similar signal is also observed between replicates. Can you please interpret these points to help understand them better, e.g. is there any significance to the fact that some genotypes have more cells "on" in glucose compared to galactose?
- It would be nice to highlight harmonious combinations in the plot in 2E.
- Not clear where GAL80.7 is in Figure 2D (reference in 3rd paragraph of page 4).
- "[GAL1 replacement] completely prevented growth and expression when GAL3 was also deleted" (bottom of page 4). This is not true in general, specifically in the Gal80#07 background (middle panel, bottom row).
- Understanding some methods requires knowing the details of the used programming language; perhaps the calls such as `cut()` and `round(x,3)` could be accompanied by a few more words what precisely these do (e.g. split into 60 equally-sized bins, rounded to 3rd decimal place, etc.)
- It was not clear if the red color in Figure 3B,C title and legend text was referring to the red points in the figures.
- Would it make sense to color the five classes ("inducible", etc.), and lightly shade the panels in 2C, 3D, and perhaps elsewhere, to quickly eyeball the signal while reading the genotypes?
- The arrow widths and types in figure 4 A are somewhat confusing - dashes and weights are not given in the legend.
- Would it make sense to distinguish the panels in the 3rd row of figure 4B a bit more? The

deleterious and harmonious configurations are very similar.

Reviewer #2:

Remarks to the Author:

The authors examine the expression of a GAL-reporter and growth rates in yeast strains that express combinations of GAL-4, -80 and -3 alleles. Based on the growth rates measured for the single allele replacements they make predictions for dual-allele combinations. While the experiments are simple, the reader can learn little from these results because the predictions are preliminary and arbitrary.

1) The primary prediction is based on a geometric model. This model seems to be realistic but it should be justified with biochemical control theory or other convincing arguments. The predictions is of low-to intermediate quality, which the authors formulate in the following way: "Overall this model explained 55% of variance across the dataset. These low values stem from the fact that pathway-level epistasis leads to constitutive and leaky expression profiles, which dominate signal in the dataset (Fig S3H). For example, all pairwise combinations of constitutive GAL80 and uninducible GAL3 are constitutive and therefore grow at high rates, when the prediction prediction is that they will grow slowly due to GAL3's low growth rate...[Further examples]".

These central findings are hidden in the Methods and the Supplemental information, even though they should be placed in the main text and figures.

It must be also emphasized that the predictions may become even worse if alleles were combined that predict intermediate growth rates, positioned between the growth rates of the two individual alleles combined in the relevant strains. However, most predictions cluster around the growth rates of that of the single alleles. Seemingly, few alleles (mutants) were constructed that predict intermediate growth for the dual-allele combinations.

2) Since the primary predictions are limited, the authors discretize and reclassify the data and make new predictions, which are better. However, these reclassifications are arbitrary and thus the new results are not predictions any more but rather accommodation. The 14 additional arbitrary parameters are not explained at all. Any new parameter should be based on physical models with measured parameters. Most parameters have been measured in the GAL system but the authors fully ignore the physical models.

Reviewer #3:

Remarks to the Author:

This manuscript reports a cohesive set of experiments that demonstrate how the quantitative complexities of even simple, well understood regulatory networks can lead to interesting, unexpected dynamic behaviors, including alternate "mutant" wirings that yield outputs similar to wild type. As such, it will be a valuable addition to the literature on epistasis and regulatory evolution. The experiments and analyses appear to have been rigorously conducted.

The authors produced a "combinatorially complete" set of tens of variants at three different loci (GAL4, GAL80, GAL3) in the GAL regulatory network (a few thousand strains total). Analysis of gene expression revealed stereotypical inducible, uninducible, constitutive or leaky dynamics of the target GAL1 promoter, suggesting that the range of genetic variation led to discrete rather than continuous variation in the phenotype. In general, single mutant dynamics predicted double mutant dynamics well, but there were exceptions, including what the authors term (after Wright) "harmonious combinations" (alternative ways to get inducible behavior). The authors show these harmonious combinations to indeed be alternative solutions, in the sense that their differences from the wild type network can be revealed by mutations in other genes (GAL3 and GAL1). These

experiments were well conceived, for example the galactokinase from other species was used to replace that function in GAL1 mutants so as to isolate the galactose-sensing role of GAL1.

Although some of the results of this manuscript are not surprising, and as the authors acknowledge (top of pg. 4) there was some prior information about combinations of GAL4 and GAL80 mutations that restore inducibility, the strength of the manuscript is in its comprehensive nature, painting a complete picture of the "phenotypic space" of the GAL regulatory network. This is an important advance and I predict it will become a key example that future work on gene interactions will refer to.

I like the representation in Fig 4 of the genotypic space and the possible evolutionary transitions. Of course, natural selection will see very subtle differences between "inducible" genotypes, and it very well might be the case that the wild type is distinct from the harmonious combinations in critical ways. But the core message of the manuscript (that the effects of mutations can depend greatly on context) nonetheless remains and is valuable.

I have some minor suggestions for improving the manuscript:

1) pg. 2 last paragraph: Fig 1B should be referred to before Fig 1C-F, or the order of the figure panels changed.

2) In Fig 1D it is not clear which data are being presented (one of the genotypes for each subplot or all of them averaged together?).

3) For GAL induction by switching from glucose to galactose medium, many people incubate with raffinose before galactose to de-repress. Can the authors comment on why they did not do that and what effect, if any, it might have on their results?

4) The left plot in Fig 1F is not very informative because the genotypes with low fraction ON in glucose are either inducible or uninducible, which as expected have very different growth rates in galactose, so the regression line and its confidence interval do not really mean that much.

5) pg. 3 first paragraph: I would have liked to see more detail on the random and directed mutagenesis here rather than having to find it in the Methods and supplement. (By what criteria were the mutants chosen?)

6) pg. 3 last paragraph: I get the gist of the analysis here, but the description is a bit confusing. It is not immediately clear what "the 23 unique locus-cluster combinations" are, and Fig 2D does not help very much because it shows at top 32 pairwise combinations (8+8+16) rather than the 36 stated below.

7) I also found Fig 2E a bit confusing. I think part of the reason is that a main point in the text (pg. 4 first paragraph) is that some combinations of constitutive GAL4 and uninducible GAL80 variants give inducible double mutants, but in Fig 2E those show up as a dark-shaded, unlabeled part of the left bar (if I am understanding correctly). Is there a different way to represent these data that brings out the key points more?

Reviewer #1 (Remarks to the Author):

In this paper, New and Lehner thoroughly dissect a model gene regulatory network by a combination of first, second, and higher order genetic perturbations. They evaluate a galactose reporter and cell growth rate in a large number of genetic backgrounds, and build models that allow predicting one from the other with good accuracy. The data and code supporting the publication are freely available. Overall, this nicely designed and reported study is an important landmark in and exemplar of the genetic variety that can underlie nominally wild type phenotypes, and demonstrates the relevance of higher order interactions.

Major comments

** Terminology. "Harmonious" combinations are also known as genetic suppression. Given the amount of confusion around terminology in discussing genetic interactions, perhaps it is appropriate not to introduce new definitions, and use definitions of masking, suppression, bypass suppression, etc. to describe the observations? I am not against the use of the word per se, but to the lack of clear link to the commonly used terms.*

The referee raises a good point. We have amended the text to include classic genetic terminology and we have more precisely defined 'harmonious combinations' (which is actually an old term used by Sewall Wright, reference 15 in the text, 1932).

"Although most double mutants exhibited highly stereotyped gene expression and predictable fitness patterns, particular combinations of mutations sometimes had very different phenotypes compared to those expected from their individual phenotypic effects. For example, 12% of pairings of Constitutive GAL4 alleles and Uninducible GAL80 variants yielded Inducible phenotypes, examples of genetic suppression (Fig S4). Similar interactions have been previously described for distinct GAL4-GAL80 allele combinations. Other suppressor combinations yielding WT-like Inducible or Leaky phenotypes included all pairings of Uninducible GAL3 with Leaky GAL80 variants (Fig 2D and Fig S4). These examples of suppression reconstitute multiple aspects of WT-like Inducibility traits, including low gene expression in glucose, high expression in galactose, and high growth rates in galactose and can be considered as examples of 'harmonious' allelic combinations: alternative genetic solutions to the core Inducible phenotype of the GAL pathway."

And in the conclusion:

"Our dataset therefore contains multiple examples of 'harmonious' combinations that exhibit the key WT-like Inducible phenotype of low expression in glucose, high expression in galactose, and high rates of growth in galactose (grey squares in Fig 4A). Four are particularly notable in their altered dependence on the GAL3 and GAL1 sensor genes ('HC' squares in Fig 4A)."*

** Mechanisms of achieving harmonious combinations of mutations. Are the harmonious mutations in the repressor Gal80 and the activator Gal4 compensatory, individually destroying, but in combination restoring a physical interaction?*

The referee has touched on an interesting topic that we agree deserves more attention in the text. We have added further discussion about the mechanism we envision that underlies the phenotype of these mutants.

In the discussion, we now say in reference to the GAL80S-1 + GAL4C combinations which restore a WT-like switch with altered dependence on GAL3 and GAL1:

“The molecular mechanism underlying the GAL4C + GAL80S-1 double mutant phenotype is intriguing. The GAL pathway is tightly repressed in glucose for GAL4C+GAL80S-1 backgrounds, suggesting that GAL80S-1 associates more tightly than WT GAL80 to GAL4C variants. Similarly, the dominant-repressive nature of the GAL80S-1 mutant is suppressed in the GAL4C backgrounds, allowing high expression in galactose. This implies that GAL80S-1 must not only bind less efficiently to Gal4Cp, but also retain the ability to interact with Gal1p and Gal3p to allow relief of repression in galactose media.

Mechanistically it is still not clear why this mutant background exhibits a higher dependency on GAL1 compared to GAL3. For example, we did not observe this in other GAL80S + GAL4C pairings. Most other GAL80S variants showed abilities to repress GAL4C, however tended to exhibit more “Constitutive” or “Uninducible” characters, with similar degrees of activation in glucose and galactose environments that did not depend on GAL1 or GAL3 (Fig S6-S7). This suggested that these mutants might bind less strongly to Gal1p and Gal3p in addition to (or instead of) tighter association to Gal4p. Mutant combinations including GAL80S-2 in combination with GAL4-L868K or GAL4-L868P exhibit partial Inducibility in galactose, with highly leaky expression in glucose but increased expression and growth in galactose that is partially dependent on GAL1 and GAL3 (Fig S6-S7). However, in contrast to the GAL4C + GAL80S-1 backgrounds, in these cases the increased expression in galactose is more dependent on GAL3 relative to GAL1 (Fig S6-S7). Given these results, we conclude that the GAL80S-1 + GAL4C backgrounds represent a special case in their increased dependency on GAL1 compared to GAL3. It seems most likely that, compared to the WT Gal80p, the Gal80S-1p variant exhibits altered binding affinity towards Gal1p in addition to Gal4p.”

** Definition of harmonious. Figure 4A depicts a range of genotype-phenotype combinations, but only some are defined as harmonious, even though the phenotype is similar to that of wild type. Specifically, why are WT+Gal1:GALK and Gal80.07 + GAL1:GALK not considered harmonious?*

The reviewer has a fair point -- all inducible backgrounds (square grey boxes in the landscape of Fig 4A) could be considered “harmonious combinations”. As we say in the figure legend, we highlight these particular combinations because they illustrate backgrounds in which the effects of the galactose sensors GAL1 and GAL3 have changed. We have reworded the main text accordingly to read:

“Our dataset therefore contains multiple examples of ‘harmonious’ combinations that exhibit the key WT-like Inducible phenotype of low expression in glucose, high expression in galactose, and high rates of growth in galactose (grey squares in Fig 4A). Four are particularly notable in their altered dependence on the GAL3 and GAL1 sensor genes (‘HC’ squares in Fig 4A).”*

And in the figure legend

“While many ‘harmonious’ combinations of WT-like phenotypes are present, we highlight particular combinations to illustrate where galactose sensing requirements have changed appreciably (HC).”*

** Phenotypic prediction. Only a few combinations of prediction are considered (growth from single mutant YFP values). It would be interesting [but not required] to also see how well single mutants predict double and higher order mutants, using both growth and YFP as predictors and targets in the four possible*

combinations. (Page 5 3rd paragraph). If the Authors choose not to, please explain why only a single direction of prediction is interesting.

We performed the final experiment to test how certain unexplained double-mutant combinations depended upon galactose sensing. The number of unique alleles that would correspond to a given gene expression class are therefore fairly limited, so if we were to perform a similar analysis as presented for the second experiment, where many alleles represented few classes, we would end up with almost as many parameters in our model as unique genotypes. Therefore we feel that such an analysis would not present a significant insight beyond the model as presented for the second pairwise mutation experiment.

Minor comments

- It would be nice to have the "leaky" and "weak" classes also annotated on Figure 2C. Otherwise, not clear e.g. where points at (0.25,0.5), (0.55,0.45), (0.65,0.75) are.

We assume the reviewer means the annotations on Fig 2B. We have changed this figure to color the double mutants with the color theme of the 5-class gene expression profiles. The single mutants, formerly colored according to their locus, are now black with shapes corresponding to locus. To reduce noise in the figure, we eliminated the error bars that were present in the previous figure.

We also noticed that generating the figure (formerly 2D, now 2F) we accidentally deleted the annotation for the Weak Expression class and have amended this in the re-submission.

- The cluster centroids are off-diagonal in figure 2B. Figure S1B (expt 2 vs expt 3) suggests that similar signal is also observed between replicates. Can you please interpret these points to help understand them better, e.g. is there any significance to the fact that some genotypes have more cells "on" in glucose compared to galactose?

This is an interesting point and we're not sure why exactly. Fig 2B shows the fraction ON vs fraction OFF, a classification determined by the proportion of cells falling above a threshold determined by the 95% percentile of autofluorescent cells. Certain data points correspond to genotypes which had a higher fraction ON in glucose compared to galactose. Most correspond to genotypes containing a GAL80S variant, primarily the GAL80.30 allele. These genotypes all have a low mean expression in glucose or galactose, so the cells contributing to the fraction ON signal all have a low expression of the GAL pathway reporter signal which falls just above the threshold for ON and OFF. In fact, the mean GAL1p-YFP signals of these clones across all cells in the population in galactose and glucose are very similar, implying that no new GAL protein is produced in the galactose environment. It could therefore be that the cells contributing to the fraction ON signal in glucose are able to grow to some extent in galactose because of the little GAL protein they are expressing, diluting the GAL signal in progeny cells, which causes them to shift from being classified as "ON" to "OFF".

In the figure above, we took the mean expression for all clones bearing the GAL80.30 allele, faceted horizontally by whether the clones exhibited a higher fraction ON in glucose than in galactose (red lines are expression in galactose, blue lines are expression in glucose). You can see that for the GAL80.30 background, all the clones with higher fraction ON in glucose than in galactose have a low mean expression values. Although we're not sure why the distributions shift like this, it could be that the GAL protein expressed by the cells in glucose is diluted to a few daughter cells in galactose, diluting the overall GAL1p-YFP signal in these cells and shifting their classification from ON to OFF.

- *It would be nice to highlight harmonious combinations in the plot in 2E.*

Highlighting these particular combinations in Fig 2E would be difficult to see. However, we agree with the spirit of this comment and generally felt that that we should illustrate which double mutant phenotypes arise from single-mutant pairings from the second experiment. To address this, and the comments of reviewer 3, we have added an additional set of supplementary figures illustrating how different mutation classes combined with one another to the supplement (now Fig S4), added a plot statistics summarizing the (limited) diversity of double-mutant phenotypes that arose from a given pair of single-mutant class pairing to Fig 2 (now Fig 2D). We first refer to this analysis in the text with this new paragraph:

"We found that only a limited number of expression phenotype classes were observed for double mutants when single mutants from particular phenotypic classes were combined (Fig 2D, Fig 2E). For example, when Leaky GAL80 mutants were combined with Uninducible GAL3 mutants, 222/234 of the double mutants (95%) fell into the Leaky phenotypic class, with the remaining 5% classified as Inducible. The single mutants in GAL3, GAL4 and GAL80 fell into two, four, and four phenotypic classes, respectively, giving a total of 32 distinct paired-locus expression class combinations in the double mutants. For 30/32 combinations (94%), >90% of the resulting double mutants fell into a single phenotypic class (Fig 2D and S4). The most diverse double mutant phenotypes were observed when combining Uninducible GAL80 mutants with Constitutive GAL4 mutants, in which case observed all five phenotypic classes were observed. However, even in this case, 52% (103/200) of the double mutants were Constitutive, a significant enrichment ($p = 2.01 \times 10^{-24}$, Chi-squared test and @Fig 2D and @Fig S4). These results suggest that gene expression in the GAL system changes in a largely stereotypical manner when combining mutations in the GALR genes, especially for loss-of-function mutations."

- *Not clear where GAL80.7 is in Figure 2D (reference in 3rd paragraph of page 4).*

We thank the reviewer for pointing out this typo. We have changed this accordingly in the text to refer to our new figure.

Now the text reads:

"For example, whereas the WT switch is completely dependent upon GAL3 (Fig 1D and Fig 3), GAL3 was not required for induction in the Leaky GAL80.07 mutant (Fig 3), nor was it required for Constitutive expression in GAL4C mutants (Fig 3D)."

- *"[GAL1 replacement] completely prevented growth and expression when GAL3 was also deleted" (bottom of page 4). This is not true in general, specifically in the Gal80#07 background (middle panel, bottom row).*

Regarding the comment “middle panel bottom row” we assume the reviewer is referring to Fig 3D, where constitutive expression was seen in all GAL80.07 backgrounds which bore the Constitutive GAL4 mutations. The quoted statement that GAL80.07 depended on GAL1 and GAL3 for expression was true for inducible GAL4 backgrounds. We have amended the text to emphasize this.

“In the Leaky GAL80.07 background, replacing GAL1 by GAL1::GALK also had no effect. However, in inducible GAL4 backgrounds, it completely prevented growth and expression when GAL3 was also deleted.”

- Understanding some methods requires knowing the details of the used programming language; perhaps the calls such as `cut()` and `round(x,3)` could be accompanied by a few more words what precisely these do (e.g. split into 60 equally-sized bins, rounded to 3rd decimal place, etc.)

We thank the reviewer for this advice. We have amended the methods text to include more straightforward language.

“The predicted FITC-A value of a PE-A reading was predicted by a linear model of FITC-A signal as a function of the correlated signal PE-A using the function $lm(\log(\text{FITC-A}) \sim \log(\text{PE}))$, and these predicted values were used in the rare cases where a cell’s FITC-A signal exceeded the machine’s maximal measurement value. Then key parameters of FITC-A distributions were extracted, including the mean YFP signal, fraction ON (the proportion of cells falling above an empirically determined cutoff based on autofluorescent cell controls). Because many lowly expressing cells gave negative values at the flow cytometer, we calculated a pseudo- \log_{10} FITC-A measurement as the \log_{10} value of the raw FITC-A plus 1000 A.U. - 3 \log_{10} arbitrary units using the function $\log_{10}(\text{raw FITC-A measurement} + 1000) - 3$. These pseudo- \log_{10} fluorescence intensity values were broken into 60 equally-spaced bins using the function `cut()`. Cells were counted in the glucose and galactose environments by determining the slope term in of events / second information of each FCS file using a linear model $lm(\text{events} \sim \text{ms})$. Glucose-grown biological replicate clones were then matched with their next-day galactose measurements. We calculated cell densities by multiplying the events/second measurement by the known sampling rates extracted from the .xml file. The density of the culture in galactose was calculated as this measured cell density parameter multiplied by the inverse of the dilution factor from glucose the day before (150/9). The \log_2 change of the culture $\text{Log}_2(\text{Density.GAL.grown.culture} / \text{Density.GLU.grown.Culture})$ was scored as the number of generations, and the growth rate parameter μ was calculated as the Malthusian growth rate parameter determined by the natural log of of the final density of the culture minus the natural log of the initial density divided by the number of hours of growth ($\log(\text{Density.GAL.grown.culture} / \text{Density.GLU.grown.Culture}) / 12 \text{ hours.}$)”

- It was not clear if the red color in Figure 3B,C title and legend text was referring to the red points in the figures.

This is indeed confusing and we thank the reviewer for pointing this out. We have changed the color of the “significant” $\text{FDR} < 0.05$ red points in the panels 3B and 3C to be black circles, and “non-significant” points to be lighter grey.

- Would it make sense to color the five classes (“inducible”, etc.), and lightly shade the panels in 2C, 3D, and perhaps elsewhere, to quickly eyeball the signal while reading the genotypes?

We have made these changes. Now, in Fig 2B we introduce the color scheme, coloring the double mutants by their expression class, and then in Fig 2C, the panels along the right side indicate the phenotype following the color scheme from Fig 2F (formerly Fig 2D), and Fig 2E the names of the bars are colored according to this scheme as well. Similarly, we have annotated each unique genotype in Fig 3D with the given combinatorial mutant's phenotype classification and corresponding color.

- The arrow widths and types in figure 4 A are somewhat confusing - dashes and weights are not given in the legend.

The arrow widths and line types are the same in Fig 4A and we are not aware of any dashes in this figure. Perhaps the orientation of the arrows makes them seem of variable widths?

- Would it make sense to distinguish the panels in the 3rd row of figure 4B a bit more? The deleterious and harmonious configurations are very similar.

The reviewer is correct that the "rewiring" illustration in this case, while phenotypically drastic, shows subtle difference between these two panel sections (key difference is in the arrow width between GAL80 and GAL4). @We have added variation to the greyscale of the arrows in addition to varying arrow widths to emphasize this difference.

Reviewer #2 (Remarks to the Author):

The authors examine the expression of a GAL-reporter and growth rates in yeast strains that express combinations of GAL-4, -80 and -3 alleles. Based on the growth rates measured for the single allele replacements they make predictions for dual-allele combinations. While the experiments are simple, the reader can learn little from these results because the predictions are preliminary and arbitrary.

We thank the referee for the comments but think that s/he has not understood the goal of our experiments and analyses. In this study we were not aiming to build a detailed mechanistic model of a system and then to use genetic perturbations to test this model. Rather, our goal was to combine together diverse mutations in different genes in a system that we know is of reasonable mechanistic complexity, and to test the extent to which standard statistical genetic models - which require no mechanistic knowledge at all - can correctly predict the phenotypic effects. We then tested whether measuring an intermediate phenotype - gene expression of one output gene in the system - could improve these predictions. We have sought to clarify this as we describe below.

1) *The primary prediction is based on a geometric model. This model seems to be realistic but it should be justified with biochemical control theory or other convincing arguments. The predictions is of low-to intermediate quality, which the authors formulate in the following way:*

"Overall this model explained 55% of variance across the dataset. These low values stem from the fact that pathway-level epistasis leads to constitutive and leaky expression profiles, which dominate signal in the dataset (Fig S3H). For example, all pairwise combinations of constitutive GAL80 and uninducible GAL3 are constitutive and therefore grow at high rates, when the prediction prediction is that they will grow slowly due to GAL3's low growth rate....[Further examples]"

These central findings are hidden in the Methods and the Supplemental information, even though they should be placed in the main text and figures.

We agree that more information about this statistical approach should be included in the main text. It is based on early population genetics theory for how the effects of new mutations should combine (Fisher 1930). To our knowledge this precedent for statistical analysis has no biophysical control theory basis - it is just the standard model used in quantitative genetics @We have added explanation of this approach and analysis to the text.

“We next asked whether the growth rate in galactose of double mutants could be predicted from the phenotypes of single mutants. We first tested the extent to which the growth rates of double mutants were predicted by simply multiplying the growth rates of the single mutants. This is the most commonly used null model for how mutations combine in both quantitative genetics and functional genomics and assumes that mutations have independent effects on growth. The multiplicative model of single mutants explained 55% of the variance in the growth rate of the 5120 double-mutants (Fig S5A, Methods). Predictive performance varied widely across different combinations of single-mutant classes with some combinations being generally poorly predicted. For example, all pairwise combinations of Constitutive GAL80 and Uninducible GAL3 were Constitutive and therefore grew at high rates, when the prediction was that they would grow slowly due to GAL3’s low growth rate. Similarly, fast-growing Constitutive GAL4 variants were predicted to grow slowly in Uninducible GAL3 backgrounds, when the double mutant remained Constitutive.”

It must be also emphasized that the predictions may become even worse if alleles were combined that predict intermediate growth rates, positioned between the growth rates of the two individual alleles combined in the relevant strains. However, most predictions cluster around the growth rates of that of the single alleles. Seemingly, few alleles (mutants) were constructed that predict intermediate growth for the dual-allele combinations.

The growth rate data for single mutants were bimodally distributed, with modes centered at high and low growth rates, as is typical of systematic mutagenesis studies, and therefore so were the predictions. To test whether the few intermediate values we observed showed any systematic bias, we classified “intermediate” prediction values and examined whether the deviations from expectation (epistasis values) exhibited a bias relative to predictions which fell closer to the two modes of high and low growth. Across double-mutant gene expression classifications, while there were dramatic differences observed in the distributions of residuals from these expectations, there did not appear to be particular bias in epistasis values for intermediate vs. high or low predicted values. Below we show a few figures illustrating these results.

Above we illustrate how single mutants were bimodally distributed between high and low growth rate values. This leads to a bimodal distribution of null expectations for growth rate.

Above we show the distribution of geometric model predictions of growth rate values. Using a mixed gaussian distribution model we fit two means and standard deviations to these data (grey lines). Samples which were 1.8 SD above the low mode and 1.8 SD below the high mode were classified as intermediate (blue). With this metric, ~5% of observations were classified as intermediate.

Above we illustrate the distributions of epistasis values faceted horizontally by the GAL locus pairing and vertically by the gene expression phenotype of the double mutants. This plot indicates that while there is a large bias in epistasis values for a given class of mutants, there is no systematic bias for epistasis values for samples with intermediate growth rate predictions compared to samples which had high or low expectations. Grey lines indicate 0 epistasis.

Similarly, the mean epistasis values indicate a lack of systematic bias in deviations of intermediate growth rate values compared to high or low values. Grey lines indicate 0 epistasis and the size of the circles corresponds to the number of mutants within each locus-pairing / gene expression class set.

2) *Since the primary predictions are limited, the authors discretize and reclassify the data and make new predictions, which are better. However, these reclassifications are arbitrary and thus the new results are not predictions any more but rather accommodation. The 14 additional arbitrary parameters are not explained at all.*

Based on gene expression distributions in glucose and galactose, we identified 3 gene expression classes (“Broad” classifications, including Uninducible, Inducible and Constitutive) and 5 classes (the 3 plus “leaky” and “weak expression” classes) which we used to characterize mutant phenotypes. To explain growth rates of double mutants, we considered unique combinations of single-mutant classifications. There are 14 possible double-mutant combinations for the 3-class scheme observed for GAL3, GAL80 and GAL4, and 23 possible combinations when GAL80’s ‘Leaky’ class and GAL4’s ‘Weak expression’ class were considered. The mean growth rate values of genotypes falling within these combinations were used to predict the resulting double-mutant phenotypes. Hopefully this clarifies the confusion over the number of parameters. We have better explained this in the results section in the revised manuscript:

“We next asked whether knowing the expression class of each single mutant allowed more accurate prediction of the growth rates of the double mutants. Specifically, we tested a model in which all the mutations in a particular gene in a given gene expression phenotypic class combine with all the mutations in a second gene in a given expression phenotypic class to give the same double mutant growth phenotype. Using five gene expression classifications for each single mutant (Inducible, Constitutive, Uninducible, Leaky and Weak Expression) gives a total of 23 double mutant classes in the dataset (e.g. Uninducible GAL3 x Constitutive GAL80). We simply used the mean growth rate of all of the genotypes falling into each of the 23 double mutant classes as the prediction of the growth rate of each double mutant in the class. So, for example, the mean of all double mutants of an Uninducible GAL3 plus a Constitutive GAL80 was used as the predicted growth rate for all double mutants whose single GAL3 variant was Inducible and whose single GAL80 variant was Constitutive. These mean values explained 89%, 90%, 91% of total growth rate variance for double-mutants in GAL3 vs. GAL4, GAL80 vs. GAL3 and GAL80 vs. GAL4 pairings, respectively (Methods). Across all double mutants, this model explained 91% of growth rate variance.

Using a coarser classification of only three single mutant expression phenotypes (Inducible, Constitutive, and Uninducible) gives 14 possible double mutant classes. Using the mean growth rate in each of these 14 classes to predict the growth of all double mutants still explains 89% of growth rate variance (Fig 2F, Fig S5B, Fig S5C, Fig S5D, Methods, and Supplementary Dataset 2).”

Indeed even if we only use a single genotype from each double mutant class as the predicted growth rate for all the genotypes in the class the predictions are still good.

“Moreover, even if we only used the mean growth rate of one random genotype from each expression class pairing, the predictions remain accurate, with a median of 86% (IQR = 84.0% - 88.0%) and 88% (IQR = 86.3% - 89.6%) of the total growth rate variance explained in the remaining >5,000 double mutants for the three and five expression class models, respectively.”

We also tested whether the locus identity was necessary for these models’ performances.

“Knowing the identity of the mutated genes is, however, useful for making accurate predictions: the three- and five-class expression models that ignore the identity of the mutated genes respectively explain 77% and 79% of the growth rate variance, with large errors for particular class combinations (Fig S5E). Thus, despite the strong epistasis in this system, simple models that capture the main ‘stereotypical’ genetic interactions between loci can accurately predict how pairs of diverse mutations interact.”

Any new parameter should be based on physical models with measured parameters. Most parameters have been measured in the GAL system but the authors fully ignore the physical models.

The referee touches on an interesting point about the types of mathematical modeling in biology (Gunawardena 2014). As explained above, our aim here was not to use a mechanistic model but to test statistical models that are agnostic to the underlying molecular mechanisms and so can also be applied to less well understood biological systems.

*There have been approaches developed to measure some of the *in vivo* biochemical parameters of the GAL network (for example by very interesting work from Becksei) but this requires an enormous amount of work and a large number of measurements to be made for any system. In contrast, statistical parameters which are blind to biochemical mechanism are common in quantitative genetics and this is what we have used to predict phenotypes for the GAL network in this manuscript. While an approach using biochemical modelling approaches to interpret the molecular mechanisms underlying GAL phenotypes in combinatorially complete and massively complex genetic libraries is great avenue for future research, this is orthogonal to and beyond the scope of our work here.*

Reviewer #3 (Remarks to the Author):

This manuscript reports a cohesive set of experiments that demonstrate how the quantitative complexities of even simple, well understood regulatory networks can lead to interesting, unexpected dynamic behaviors, including alternate "mutant" wirings that yield outputs similar to wild type. As such, it will be a valuable addition to the literature on epistasis and regulatory evolution. The experiments and analyses appear to have been rigorously conducted.

The authors produced a "combinatorially complete" set of tens of variants at three different loci (GAL4, GAL80, GAL3) in the GAL regulatory network (a few thousand strains total). Analysis of gene expression revealed stereotypical inducible, uninducible, constitutive or leaky dynamics of the target GAL1 promoter, suggesting that the range of genetic variation led to discrete rather than continuous variation in the phenotype. In general, single mutant dynamics predicted double mutant dynamics well, but there were exceptions, including what the authors term (after Wright) "harmonious combinations" (alternative ways to get inducible behavior). The authors show these harmonious combinations to indeed be alternative solutions, in the sense that their differences from the wild type network can be revealed by mutations in other genes (GAL3 and GAL1). These experiments were well conceived, for example the galactokinase from other species was used to replace that function in GAL1 mutants so as to isolate the galactose-sensing role of GAL1.

Although some of the results of this manuscript are not surprising, and as the authors acknowledge (top of pg. 4) there was some prior information about combinations of GAL4 and GAL80 mutations that restore

inducibility, the strength of the manuscript is in its comprehensive nature, painting a complete picture of the "phenotypic space" of the GAL regulatory network. This is an important advance and I predict it will become a key example that future work on gene interactions will refer to.

I like the representation in Fig 4 of the genotypic space and the possible evolutionary transitions. Of course, natural selection will see very subtle differences between "inducible" genotypes, and it very well might be the case that the wild type is distinct from the harmonious combinations in critical ways. But the core message of the manuscript (that the effects of mutations can depend greatly on context) nonetheless remains and is valuable.

We thank the reviewer for these comments, which we feel have captured the shortcomings, strengths and main message of our manuscript well.

I have some minor suggestions for improving the manuscript:

1) *pg. 2 last paragraph: Fig 1B should be referred to before Fig 1C-F, or the order of the figure panels changed.*

We have changed the manuscript to refer to both figure panels.

2) *In Fig 1D it is not clear which data are being presented (one of the genotypes for each subplot or all of them averaged together?).*

We have changed the text to emphasize that the mutants falling into these classes were all averaged together.

Now the figure legend now reads:

"Three phenotypic classes of GALR deletion mutants. Lines and shading are the mean and ± 1 SD of genotypes falling into the given expression class (N=4 for N=2 independent transformations for each unique genotype)."

3) *For GAL induction by switching from glucose to galactose medium, many people incubate with raffinose before galactose to de-repress. Can the authors comment on why they did not do that and what effect, if any, it might have on their results?*

The reviewer has a good point. There are a number of reasons that we chose to not use raffinose in particular.

1) Raffinose is a trisaccharide which is hydrolyzed by invertase to fermentable carbon source. Because invertase is secreted, the growth rate of a culture in raffinose depends upon its cell density (Koschwanez, Foster, and Murray 2011). A key aspect of our protocol was to dilute the culture many-fold to allow depletion of non-transformed cells, and the low culture densities required for this would have led to starvation conditions for our cultures prior to their inoculation.

2) Raffinose effectively serves as a fructose carbon source because invertase hydrolyzes it to fructose and the poorly metabolized disaccharide melibiose. Using glucose seemed reasonable for this reason, given that this is a more standard laboratory carbon source compared to fructose.

3) Finally, melibiose is a disaccharide containing glucose and galactose, and while the laboratory strain of *S. cerevisiae* weakly metabolizes this sugar, other *S. cerevisiae* strains (bearing the highly intraspecifically variable *MEL* family of genes) and ascomycete species are known to do so, and therefore our results would not translate to these other systems. Further, should any of our mutants have stimulated the formation of

melibiose hydrolysis (for example, the *GALK* paralogs from *C. albicans* or *E. coli*), or if a given batch exhibited chemical degradation, the free galactose might auto-induce the pathway.

We decided to use cultures coming from saturated low-concentration glucose cultures as the “glucose” environment. Having depleted glucose, these cultures were in the process of respiration and entering stationary phase and therefore the GAL pathway was “derepressed” allowing growth in galactose. The timing of the entry into this phase of nutrient limited growth was tightly controlled by initial inoculation density and the period of time the cells were allowed to grow, leading to high reproducibility between experiments. We have amended the text to reflect this rationale. Now in the section where we discuss the experimental setup, we say:

“The first environment was an initial “uninducing” condition where cells had reached saturation in media with a low concentration of glucose, and the second condition was “inducing” after 12 hours of growth in galactose.”

4) The left plot in Fig 1F is not very informative because the genotypes with low fraction ON in glucose are either inducible or uninducible, which as expected have very different growth rates in galactose, so the regression line and its confidence interval do not really mean that much.

This is true and was our intention. A statistical genetic analysis which is blind to molecular mechanism could include a regression such as this, and would correctly conclude that GAL pathway expression in glucose is related to growth in galactose.

5) pg. 3 first paragraph: I would have liked to see more detail on the random and directed mutagenesis here rather than having to find it in the Methods and supplement. (By what criteria were the mutants chosen?)

We have included more discussion of why mutants were chosen in the main text. Now, before describing the second experiment, we say:

“Next, to investigate how richer spectra of mutations in the three GALR genes interact, we used PCR mutagenesis to generate variants of the three GALR genes and phenotyped them using flow cytometry (Supplementary Tables 1-3 and Methods). We observed GALR alleles that individually behaved as detrimental, mildly detrimental, and WT-like. We also observed mutations in GAL80 (“GAL80S”) that led to Uninducible phenotypes, as well as mutations in GAL4 (“GAL4C”) that caused Constitutive expression. Since these were relatively rare, we generated a number of previously described gain-of-function mutants in GAL80 and we mutagenized position L868 of GAL4, a site in the GAL80-binding interface of the GAL4 activation domain where mutation to P had previously been shown to drive a constitutive phenotype (Methods and Supplementary Tables S1-S3).”

6) pg. 3 last paragraph: I get the gist of the analysis here, but the description is a bit confusing. It is not immediately clear what “the 23 unique locus-cluster combinations” are, and Fig 2D does not help very much because it shows at top 32 pairwise combinations (8+8+16) rather than the 36 stated below.

Note that Fig 2D is now Fig 2F. We thank the reviewer for pointing out this typo in the figure: the true number of possible combinations was 32. We have changed this in the figure.

7) I also found Fig 2E a bit confusing. I think part of the reason is that a main point in the text (pg. 4 first paragraph) is that some combinations of constitutive GAL4 and uninducible GAL80 variants give inducible double mutants, but in Fig 2E those show up as a dark-shaded, unlabeled part of the left bar (if I am

understanding correctly). Is there a different way to represent these data that brings out the key points more?

We have made a new supplementary figure Fig S4 and main figure Fig 2D showing which double-mutant phenotypic classes arose from underlying single-mutant classes. Hopefully this helps the reviewer to better understand the text and double-mutant outcomes from single mutants.

Summary of changes to figures

Please note that we have made the following changes to the figures. We have added an additional supplementary figure illustrating how different mutation classes combined with one another (now Fig S4), added a plot of statistics summarizing the diversity of double-mutant phenotypes that arose from a given pair of single-mutant class pairings to Fig 2 (now Fig 2D). In adding Fig S4, we split Fig S3 into two sections: the first sections (Fig S3A-D) describing the classifications of different mutants based on expression characteristics, and the second (Formerly Fig S3E-H, now Fig S5A-D) illustrating the predictive modeling results. We also added another analysis to the predictive modeling section, and this plot is now Fig S5E. We have also re-arranged the order of the predictive modeling section, first presenting the geometric model, followed by the classification models. We split the supplementary figures relating to these models into a new supplementary figure.

In summary, the current figures in this resubmission are now (Old Fig → New Fig):

Fig 2:

With a new figure Fig 2D

Fig 2D → 2F

Fig S3:

Figures Fig 3A-D (the same as before) are the only figures in Fig S3 now.

Fig S4:

A new figure Fig S4 has been added.

Fig S5:

Fig S3H → S5A

Fig S3E-G → S5B-D

With a new figure Fig S5E being added.

With the addition of two supplementary figures, we increased the numbering of the remaining three figures +2.

Fig S6:

Fig S4 → Fig S6

Fig S7

Fig S5 → Fig S7

Fig S8

Fig S6 → Fig S8

References

Acar, Murat, Attila Becskei, and Alexander van Oudenaarden. 2005. "Enhancement of Cellular Memory by Reducing Stochastic Transitions." *Nature* 435 (7039): 228–32.

- Bennett, Matthew R., Wyming Lee Pang, Natalie A. Ostroff, Bridget L. Baumgartner, Sujata Nayak, Lev S. Tsimring, and Jeff Hasty. 2008. "Metabolic Gene Regulation in a Dynamically Changing Environment." *Nature* 454 (7208): 1119–22.
- Fisher, Ronald Aylmer. 1930. "The Genetical Theory of Natural Selection." <https://doi.org/10.5962/bhl.title.27468>.
- Gunawardena, Jeremy. 2014. "Models in Biology: 'accurate Descriptions of Our Pathetic Thinking.'" *BMC Biology*. <https://doi.org/10.1186/1741-7007-12-29>.
- Koschwanez, John H., Kevin R. Foster, and Andrew W. Murray. 2011. "Sucrose Utilization in Budding Yeast as a Model for the Origin of Undifferentiated Multicellularity." *PLoS Biology* 9 (8): e1001122.
- Venturelli, Ophelia S., Hana El-Samad, and Richard M. Murray. 2012. "Synergistic Dual Positive Feedback Loops Established by Molecular Sequestration Generate Robust Bimodal Response." *Proceedings of the National Academy of Sciences of the United States of America* 109 (48): E3324–33.

Reviewers' Comments:

Reviewer #1:

Remarks to the Author:

The authors have addressed all my concerns. Congratulations on a nice paper.

Reviewer #2:

Remarks to the Author:

The authors declined including a mechanistic model arguing that it is difficult and that they focus on statistical models agnostic of mechanisms. Applications of statistical models (e.g. statistical genetics) can be very useful provided they are used appropriately (Minor comment 1). On the other hand, it is simply not true that authors avoid mechanistic argumentation. In fact, the manuscript is littered with qualitative mechanistic argumentation, which is vaguely and incorrectly linked to the quantitative statistical analysis of gene expression and growth rates (Major comment 1).

Major comment 1

Despite the authors' statement that they ignore molecular mechanisms, the authors argue mechanistically both in the design and interpretation phases of the study. For example, "This would be consistent with GAL4C single-mutant variants being de-repressed in glucose not only because of reduced GAL80 repression (Fig 3D), but also due to positive feedback via GAL1, which, through a baseline level of leaky expression (Fig 3D) and ability to repress GAL80(9), could reach sufficient abundance to constitutively activate the system"

In this formulation, the sentence is even simply wrong. The capital italic notation refers to genes. The GAL4 (or GAL4C) gene is not de-repressed in glucose at all. On the contrary, the expression of GAL4 is repressed by glucose, mediated by Mig1 binding sites in Gal4 promoter (ref: Johnston et al, 1994).

Furthermore, the feedback through GAL1 is highly context-dependent and may have hardly any effect in some genetic backgrounds but a very strong effect in other gene backgrounds. How do the authors know that the positive feedback is relevant in the above cell background?

It is not even clear if the authors refer to GAL1::GALK or GAL1. If they refer to GAL1::GALK, positive feedback cannot even arise through signaling. The authors refer to reference 9 but that reference does not mention at all the term "positive feedback" loop, which is a clear case of improper citation practice. Gal1 can in principle act also as a negative feedback because it reduces the galactose concentration enzymatically and thus it deactivates the galactose signal. Why do the authors omit to cite a paper that identified a positive feedback loop through Gal1p?

A positive feedback is a notion coming mechanistic models and not from statistical models. The authors ignore a whole range of findings in this field and engage in a vague and flawed argumentation.

The cited sentence refers to interaction of four different mechanisms but formulated in a qualitative way. In the light above the above sentence, the authors' statement "as explained above, our aim here was not to use a mechanistic model but to test statistical models that are agnostic to the underlying molecular mechanisms" is misleading or simply wrong. If the authors' study is agnostic of molecular mechanisms why is the summary figure (Figure 4B) full of mechanistic details (sensor-effect, positive feedback loop,...).

I did not ask the authors to measure molecular kinetic parameters of proteins, metabolites and RNA in the system since many are already available. They just have to adapt to their main findings. It is not clear why the author engage in reasoning on complex molecular mechanisms but decline to do it consistently and precisely with mechanistic models.

Minor comment 1

For example, it is very useful to apply gene mapping to identify genes in a given process.

However, when the genes are already identified then it makes little sense to use gene mapping.

The GAL networks is thoroughly studied mechanistically, but the authors ignore that. Therefore, it makes little sense to use a pure statistical approach when mechanistic details are available. Epistasis (including higher order epistasis) is not surprising at all in a network with well known nonlinear kinetics. It would have been surprising if there was no higher order epistasis detected. Statistical methods make sense as long as they are combined with mechanistic models, at this stage of research.

Minor comment 2

The prediction at intermediate growth rate should have been analyzed with residual analysis. Most predicted growth rates cluster to two extremes: slow (around 0.1 h⁻¹) and fast growth rates (around 0.2 h⁻¹). This is simply due to the fact that the majority of mutants are determined by dominant effects ("For example, all pairwise combinations of constitutive GAL80 and uninducible GAL3 are constitutive and therefore grow at high rates, when the prediction prediction is that they will grow slowly due to GAL3's low growth rate"). As the authors state, 95% of double mutants result in slow or fast rates, and only 5% of double mutants result at intermediate growth rates. This poses very little demand on the predictions and the seemingly good predictions are simply due to the fact that most combinations of mutations cluster to these extremes, which automatically guaranties high R² values. It is clear that predictions are not good when the predicted values are intermediate, as in the case in Figure 2D, right panel (GAL80, GAL4), predicted growth rate around 0.15 : at this predicted intermediate growth, there is a broad range of measured growth rates, revealing the limitations of the predictions. Yet, the overall R² is high.

Reviewer #3:

Remarks to the Author:

The authors have responded well to all critiques of the original manuscript. This paper will make an excellent addition to the literature on genetic interactions.

Aaron New and Ben Lehner Response to Reviewer 2 and Editorial requests.

The editor has requested:

Based on these comments [Reviewer #3's arbitration] we have taken the decision to overrule Reviewer #2 and to accept your manuscript. Nevertheless, we invite you to revise your paper one last time to include/discuss that Mig1 represses GAL4 in glucose (major comment 1 reviewer #2) and acknowledge that the prediction accuracy is not high for the intermediate growth rates (minor comment2 reviewer #2).

Below we have responded to these requests:

Reviewer #2 (Remarks to the Author):

The authors declined including a mechanistic model arguing that it is difficult and that they focus on statistical models agnostic of mechanisms. Applications of statistical models (e.g. statistical genetics) can be very useful provided they are used appropriately (Minor comment 1). On the other hand, it is simply not true that authors avoid mechanistic argumentation. In fact, the manuscript is littered with qualitative mechanistic argumentation, which is vaguely and incorrectly linked to the quantitative statistical analysis of gene expression and growth rates (Major comment 1).

Major comment 1

Despite the authors' statement that they ignore molecular mechanisms, the authors argue mechanistically both in the design and interpretation phases of the study. For example,

"This would be consistent with GAL4C single-mutant variants being de-repressed in glucose not only because of reduced GAL80 repression (Fig 3D), but also due to positive feedback via GAL1, which, through a baseline level of leaky expression (Fig 3D) and ability to repress GAL80(9), could reach sufficient abundance to constitutively activate the system"

In this formulation, the sentence is even simply wrong. The capital italic notation refers to genes. The GAL4 (or GAL4C) gene is not de-repressed in glucose at all. On the contrary, the expression of GAL4 is repressed by glucose, mediated by Mig1 binding sites in Gal4 promoter (ref: Johnston et al, 1994).

Reviewer #2 is mistaken here. Johnston 1994 (Johnston, Flick, and Pexton 1994) did not measure *GAL4* expression, rather they used a mutant in *GAL4* which is "derepressed" in glucose. Importantly, this variant did not have a phenotypic effect on GAL pathway induction kinetics, implying that glucose repression of *GAL4* plays a minimal role in pathway induction.

Earlier, others observed that *GAL4* expression is affected by *MIG1* repression (Nehlin, Carlberg, and Ronne 1991) and that Mig1p binds in the promoter of *GAL4*. To our knowledge this observation has not been replicated, and its phenotypic effect is contradicted by Johnson 1994, as well as more recent highly quantitative analyses in the standard laboratory strain, all of which conclude that *GAL4* is constitutively expressed at the transcript and protein level (see e.g. (Ghaemmaghani et al. 2003)). In this latter study, Gal4p is present in glucose at 6.6 nM or 166 copies per cell, in par with parameter estimates from dynamical modeling studies (e.g. (Gencoglu, Schmidt, and Becskei 2017)). Further, we know that this concentration of glucose is sufficient to allow high pathway induction in glucose + galactose mixtures (Escalante-Chong et al. 2015). Finally our own results contradict this reasoning: in Δ GAL80 / "Constitutive" *GAL80* backgrounds, the GAL pathway is highly induced in glucose, and this expression is dependent on *GAL4* (Δ GAL80 Δ GAL4 strain does not express the GAL pathway). This implies that *GAL4* is expressed sufficiently to induce the GAL pathway in the presence of glucose.

Given our evidence, and those of others, we have opted to not discuss the effect of MIG1 on GAL4 expression. However, we have inserted the following into the text, which could have led to some of the confusion above:

"This would be consistent with the GAL pathway of GAL4C single-mutant variants being de-repressed in glucose not only because of reduced GAL80 repression (Fig 3D), but also due to positive feedback via GAL1, which, through a baseline level of leaky expression (Fig 3D) and ability to repress GAL80(9), could reach sufficient abundance to constitutively activate the system"

Furthermore, the feedback through GAL1 is highly context-dependent and may have hardly any effect in some genetic backgrounds but a very strong effect in other gene backgrounds. How do the authors know that the positive feedback is relevant in the above cell background?

It is not even clear if the authors refer to GAL1::GALK or GAL1.

Simply reading the quotation pasted by reviewer, one can see that we are talking about "GAL1" giving positive feedback. As the reviewer points out below, we intentionally knocked out this feedback using the GAL1::GALK construct.

If they refer to GAL1::GALK, positive feedback cannot even arise through signaling. The authors refer to reference 9 but that reference does not mention at all the term "positive feedback" loop, which is a clear case of improper citation practice. Gal1 can in principle act also as a negative feedback because it reduces the galactose concentration enzymatically and thus it deactivates the galactose signal. Why do the authors omit to cite a paper that identified a positive feedback loop through Gal1p? A positive feedback is a notion coming mechanistic models and not from statistical models. The authors ignore a whole range of findings in this field and engage in a vague and flawed argumentation.

The cited sentence refers to interaction of four different mechanisms but formulated in a qualitative way. In the light above the above sentence, the authors' statement "as explained above, our aim here was not to use a mechanistic model but to test statistical models that are agnostic to the underlying molecular mechanisms" is misleading or simply wrong. If the authors' study is agnostic of molecular mechanisms why is the summary figure (Figure 4B) full of mechanistic details (sensor-effect, positive feedback loop,...).

I did not ask the authors to measure molecular kinetic parameters of proteins, metabolites and RNA in the system since many are already available. They just have to adapt to their main findings. It is not clear why the author engage in reasoning on complex molecular mechanisms but decline to do it consistently and precisely with mechanistic models.

Our citation to reference 9 is in regard to the ability of Gal1p to bind to Gal80p even in the absence of galactose. What we speculated about was a novel mechanism of positive feedback whereby GAL1 auto-induces the pathway in the absence of galactose. To our knowledge this mechanism has not been described for the GAL pathway. To confirm this mechanism would have required considerable extra work, including measurement of "molecular kinetic parameters of proteins metabolites and RNA" for these novel alleles, beyond the scope of the current manuscript. It is a discussion point and a hypothesis to be tested in future work.

Minor comment 1

For example, it is very useful to apply gene mapping to identify genes in a given process. However, when the genes are already identified then it makes little sense to use gene mapping. The GAL networks is thoroughly studied mechanistically, but the authors ignore that. Therefore, it makes little sense to use a pure statistical approach when mechanistic details are available. Epistasis (including higher order epistasis) is not surprising at all in a network with well known nonlinear kinetics. It would have been surprising if there

was no higher order epistasis detected. Statistical methods make sense as long as they are combined with mechanistic models, at this stage of research.

Minor comment 2

The prediction at intermediate growth rate should have been analyzed with residual analysis. Most predicted growth rates cluster to two extremes: slow (around 0.1 h⁻¹) and fast growth rates (around 0.2 h⁻¹). This is simply due to the fact that the majority of mutants are determined by dominant effects (“For example, all pairwise combinations of constitutive GAL80 and uninducible GAL3 are constitutive and therefore grow at high rates, when the prediction prediction is that they will grow slowly due to GAL3’s low growth rate”). As the authors state, 95% of double mutants result in slow or fast rates, and only 5% of double mutants result at intermediate growth rates. This poses very little demand on the predictions and the seemingly good predictions are simply due to the fact that most combinations of mutations cluster to these extremes, which automatically guarantees high R² values. It is clear that predictions are not good when the predicted values are intermediate, as in the case in Figure 2D, right panel (GAL80, GAL4), predicted growth rate around 0.15 : at this predicted intermediate growth, there is a broad range of measured growth rates, revealing the limitations of the predictions. Yet, the overall R² is high.

The reviewer refers to combinations of GAL4C with GAL80S variants which, as we point out many times in the text, give rise to the least predictable double mutant phenotypes, and inspired the third experiment in our paper.

We have amended the text again to reflect this more explicitly when discussing the result of the modeling.

“While highly predictive overall, double-mutants arising from pairings between Constitutive GAL4 alleles with Uninducible GAL80 alleles were the least predictable in their resulting growth rates (Fig 2D, points with growth predictions at ~0.15 hr⁻¹) and gene expression classes (Supplementary Fig 4).”

References

- Escalante-Chong, Renan, Yonatan Savir, Sean M. Carroll, John B. Ingraham, Jue Wang, Christopher J. Marx, and Michael Springer. 2015. “Galactose Metabolic Genes in Yeast Respond to a Ratio of Galactose and Glucose.” *Proceedings of the National Academy of Sciences* 112 (5): 1636–41.
- Gencoglu, Mumun, Alexander Schmidt, and Attila Becskei. 2017. “Measurement of In Vivo Protein Binding Affinities in a Signaling Network with Mass Spectrometry.” *ACS Synthetic Biology* 6 (7): 1305–14.
- Ghaemmaghani, Sina, Won-Ki Huh, Kiowa Bower, Russell W. Howson, Archana Belle, Noah Dephoure, Erin K. O’Shea, and Jonathan S. Weissman. 2003. “Global Analysis of Protein Expression in Yeast.” *Nature* 425 (6959): 737–41.
- Johnston, M., J. S. Flick, and T. Pexton. 1994. “Multiple Mechanisms Provide Rapid and Stringent Glucose Repression of GAL Gene Expression in *Saccharomyces Cerevisiae*.” *Molecular and Cellular Biology* 14 (6): 3834–41.
- Nehlin, J. O., M. Carlberg, and H. Ronne. 1991. “Control of Yeast GAL Genes by MIG1 Repressor: A Transcriptional Cascade in the Glucose Response.” *The EMBO Journal* 10 (11): 3373–77.